# Quantitative Understanding of VAE as a Non-linearly Scaled Isometric Embedding

## Abstract

Variational autoencoder (VAE) estimates the posterior parameters (mean and variance) of latent variables corresponding to each input data. While it is used for many tasks, the transparency of the model is still an underlying issue. This paper provides a quantitative understanding of VAE property by interpreting VAE as a non-linearly scaled isometric embedding. According to the Rate-distortion theory, the optimal transform coding is achieved by using a PCA-like orthonormal transform where the transform space is isometric to the input. From this analogy, we show theoretically and experimentally that VAE can be mapped to an implicit isometric embedding with a scale factor derived from the posterior parameter. As a result, we can estimate the data probabilities in the input space from the prior, loss metrics, and corresponding posterior parameters. In addition, the quantitative importance of each latent variable can be evaluated like the eigenvalue of PCA.

## 1 Introduction

Variational autoencoder (VAE) (Kingma & Welling, 2014) is one of the most successful generative models, estimating posterior parameters of latent variables for each input data. In VAE, the latent representation is obtained by maximizing an evidence lower bound (ELBO). A number of studies (Higgins et al., 2017; Kim & Mnih, 2018; Lopez et al., 2018; Chen et al., 2018; Locatello et al., 2019; Alemi et al., 2018; Rolínek et al., 2019) have tried to reveal the property of latent variables. However, quantitative behavior of VAE is still not well clarified. For example, there has not been a theoretical formulation of the reconstruction loss and KL divergence in ELBO after optimization. More specifically, although the conditional distribution $p_\theta(x|z)$ in the reconstruction loss of ELBO is predetermined such as the Gaussian or Bernoulli distributions, it has not been discussed well whether the true conditional distribution after optimization matches the predetermined distribution.

Rate-distortion (RD) theory (Berger, 1971), which is an important part of Shannon information theory and successfully applied to image compression, quantitatively formulates the RD trade-off optimum in lossy compression. To realize a quantitative data analysis, Rate-distortion (RD) theory based autoencoder, RaDOGAGA (Kato et al., 2020), has been proposed with isometric embedding (Han & Hong, 2006) where the distance between arbitrary two points of input space in a given metrics is always the same as L2 distance in the isometric embedding space. In this paper, by mapping VAE latent space to an implicit isometric space like RaDOGAGA on variable-by-variable basis and analysing VAE quantitatively as a well-examined lossy compression, we thoroughly clarify the quantitative properties of VAE theoretically and experimentally as follows.

1) Implicit isometric embedding is derived in the loss metric defined space such that the entropy of data representation becomes minimum. A scaling factor between the VAE latent space and implicit isometric space is formulated by the posterior for each input. In the case of $\beta$-VAE, the posterior variance of each dimensional component in the implicit isometric embedding space is a constant $\beta/2$, which is analogous to the rate-distortion optimal of transform coding in RD theory. As a result, the reconstruction loss and KL divergence in ELBO can be quantitatively formulated.
2) From these properties, VAE can provide a practical quantitative analysis of input data. First, the data probabilities in the input space can be estimated from the prior, loss metric, and posterior parameters. In addition, the quantitative importance of each latent variable, analogous to the eigenvalue of PCA, can be evaluated from the posterior variance of VAE.

This work will lead the information theoretic generative models in the right direction.

## 2 RELATED WORKS

### 2.1 VARIATIONAL AUTOENCODER AND THEORETICAL ANALYSIS

In VAE, ELBO is maximized instead of maximizing the log-likelihood directly. Let $\boldsymbol{x} \in \mathbb{R}^m$ be a point in a dataset. The original VAE model consists of a latent variable with fixed prior $\boldsymbol{z} \sim p(\boldsymbol{z}) = \mathcal{N}(\boldsymbol{z}; 0, \boldsymbol{I}_n) \in \mathbb{R}^n$, a parametric encoder $\mathrm{Enc}_\phi : \boldsymbol{x} \Rightarrow \boldsymbol{z}$, and a parametric decoder $\mathrm{Dec}_\theta : \boldsymbol{z} \Rightarrow \hat{\boldsymbol{x}}$. In the encoder, $q_\phi(\boldsymbol{z}|\boldsymbol{x}) = \mathcal{N}(\boldsymbol{z}; \boldsymbol{\mu}_{(\boldsymbol{x})}, \boldsymbol{\sigma}_{(\boldsymbol{x})})$ is provided by estimating parameters $\boldsymbol{\mu}_{(\boldsymbol{x})}$ and $\boldsymbol{\sigma}_{(\boldsymbol{x})}$. Let $L_{\boldsymbol{x}}$ be a local cost at data $\boldsymbol{x}$. Then, ELBO is described by

$$\mathrm{ELBO} = E_{\boldsymbol{x} \sim p(\boldsymbol{x})} \left[ E_{\boldsymbol{z} \sim q_\phi(\boldsymbol{z}|\boldsymbol{x})}[\log p_\theta(\boldsymbol{x}|\boldsymbol{z})] - D_{\mathrm{KL}}(q_\phi(\boldsymbol{z}|\boldsymbol{x})\|p(z)) \right]. \tag{1}$$

In $E_{\boldsymbol{x} \sim p(\boldsymbol{x})}[\,\cdot\,]$, the first term $E_{\boldsymbol{z} \sim q_\phi(\boldsymbol{z}|\boldsymbol{x})}[\,\cdot\,]$ is called the reconstruction loss. The second term $D_{\mathrm{KL}}(\cdot)$ is a Kullback–Leibler (KL) divergence. Let $\mu_{j(\boldsymbol{x})}, \sigma_{j(\boldsymbol{x})}$, and $D_{\mathrm{KL}j(x)}$ be $j$-th dimensional values of $\boldsymbol{\mu}_{(\boldsymbol{x})}, \boldsymbol{\sigma}_{(\boldsymbol{x})}$, and KL divergence. Then $D_{\mathrm{KL}}(\cdot)$ is derived as:

$$D_{\mathrm{KL}}(\cdot) = \sum_{j=1}^{n} D_{\mathrm{KL}j(x)}, \quad \text{where} \quad D_{\mathrm{KL}j(x)} = \frac{1}{2}\left(\mu_{j(\boldsymbol{x})}{}^2 + \sigma_{j(\boldsymbol{x})}{}^2 - \log \sigma_{j(\boldsymbol{x})}{}^2 - 1\right). \tag{2}$$

$D(\boldsymbol{x}, \hat{\boldsymbol{x}})$ denotes a metric such as sum square error (SSE) and binary cross-entropy (BCE) as log-likelihoods of Gaussian and Bernoulli distributions, respectively. In training VAE, the next objective is used instead of Eq. 1, where $\beta$ is a parameter to control the trade-off (Higgins et al., 2017).

$$L_{\boldsymbol{x}} = E_{\boldsymbol{z} \sim q_\phi(\boldsymbol{z}|\boldsymbol{x})}[D(\boldsymbol{x}, \hat{\boldsymbol{x}})] + \beta D_{\mathrm{KL}}(\cdot). \tag{3}$$

However, it has not been fully discussed whether the true conditional distribution matches the pre-determined distribution, or how the value of KL divergence is derived after training.

There have been several studies to analyse VAE theoretically. Alemi et al. (2018) introduced the RD trade-off based on the information-theoretic framework to analyse $\beta$-VAE. However, they did not clarify the quantitative property after optimization. Dai et al. (2018) showed that VAE restricted as a linear transform can be considered as a robust PCA. However, their model has a limitation for the analysis on each latent variable basis because of the linearity assumption. Rolínek et al. (2019) showed that the Jacobian matrix of VAE at each latent variable is orthogonal, which makes latent variables disentangled implicitly. However, they do not uncover the orthonormality and quantitative properties because they simplify KL divergence as a constant. Dai & Wipf (2019) also showed that the expected rate of VAE for the $r$-dimensional manifold is close to $-(r/2)\log\gamma + O(1)$ at $\gamma \to 0$ when $p_\theta(\hat{\boldsymbol{x}}|\boldsymbol{x}) = \mathcal{N}(\hat{\boldsymbol{x}}; \boldsymbol{x}, \gamma I_m)$ holds. The remaining challenge is to clearly figure out what latent space is obtained at a given dataset, a loss metric, and $\beta$ in the model.

### 2.2 RATE-DISTORTION THEORY, TRANSFORM CODING, AND ISOMETRIC EMBEDDING

RD theory (Berger, 1971) formulated the optimal transform coding (Goyal, 2001) for the Gaussian source with square error metric as follows. Let $\boldsymbol{x} \in \mathbb{R}^m$ be a point in a dataset. First, the data are transformed deterministically with the orthonormal transform (orthogonal and unit norm) such as Karhunen-Loève transform (KLT) (Rao & Yip, 2000). Let $\boldsymbol{z} \in \mathbb{R}^m$ be a point transformed from $\boldsymbol{x}$. Then, $\boldsymbol{z}$ is entropy-coded by allowing equivalent stochastic distortion (or posterior with constant variance) in each dimension. A lower bound of a rate $R$ at a distortion $D$ is denoted by $R(D)$. The derivation of $R(D)$ is as follows. Let $z_j$ be the $j$-th dimensional component of $\boldsymbol{z}$ and $\sigma_{zj}{}^2$ be the variance of $z_j$ in a dataset. It is noted that $\sigma_{zj}{}^2$ is the equivalent to eigenvalues of PCA for the dataset. Let $d$ be a distortion equally allowed in each dimensional channel. At the optimal condition, the distortion $D_{\mathrm{opt}}$ and rate $R_{\mathrm{opt}}$ on the curve $R(D)$ is calculated as a function of $d$:

$$R_{\mathrm{opt}} = \frac{1}{2}\sum_{j=1}^{m} \max(\log(\sigma_{zj}{}^2/d), 0), \quad D_{\mathrm{opt}} = \sum_{j=1}^{m} \min(d, \sigma_{zj}{}^2). \tag{4}$$

The simplest way to allow equivalent distortion is to use a uniform quantization (Goyal, 2001). Let $T$ be a quantization step, and $\mathrm{round}(\cdot)$ be a round function. Quantized value $\hat{z_j}$ is derived as $kT$, where $k = \mathrm{round}(z_j/T)$. Then, $d$ is approximated by $T^2/12$ as explained in Appendix H.1.

To practically achieve the best RD trade-off in image compression, rate-distortion optimization (RDO) has also been widely used (Sullivan & Wiegand, 1998). In RDO, the best trade-off is

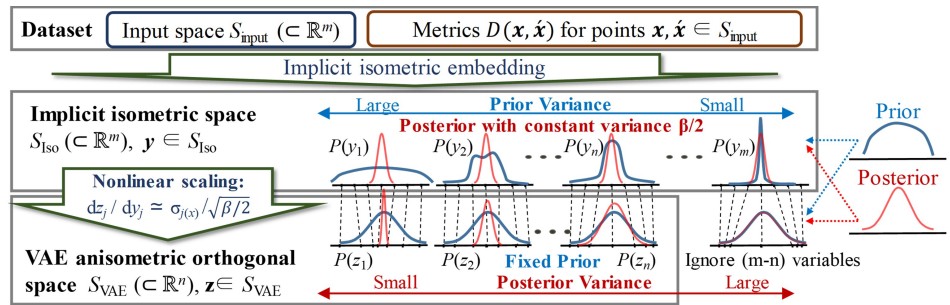

Figure 1: Mapping of VAE to implicit isometric embedding.

achieved by finding a encoding parameter that minimizes a cost $L = D + \lambda R$ at given Lagrange parameter $\lambda$. Recently, deep image compression (Ballé et al., 2018) has been proposed. In these works, instead of an orthonormal transform with sum square error (SSE) metric in the conventional lossy compression, a deep autoencoder is trained with flexible metrics, such as structural similarity (SSIM) (Wang et al., 2001) for RDO. Recently, an isometric autoencoder, RaDOGAGA (Kato et al., 2020) was proposed based on Ballé et al. (2018). They proved that the latent space to be isometric to the input space if the model is trained by RDO using a parametric prior and posterior with constant variance. By contrast, VAE uses a fixed prior with a variable posterior. In section 3, we explain that VAE can be quantitatively understood as the rate-distortion optimum as in Eq. 4 by mapping VAE latent space to implicit isometric embedding on a variable-to-variable basis as in Fig. 1 .

## 3 UNDERSTANDING OF VAE AS A SCALED ISOMETRIC EMBEDDING

This section shows the quantitative understanding of VAE. First, we present the hypothesis of mapping VAE latent space to an implicit isometric embedding space. Second, we reformulate the objective of $\beta$-VAE for easy analysis. Third, we prove the hypothesis from the minimum condition of the objective. Then, we show that ELBO can be interpreted as an optimized RDO cost of transform coding where the quantitative properties are well clarified, as well as discuss and correct some prior theoretical studies. Lastly, we explain the quantitative properties of VAE to validate the theory including approximations and provide a practical data analysis.

### 3.1 HYPOTHESIS OF MAPPING VAE TO THE IMPLICIT ORTHONORMAL TRANSFORM

Figure 1 shows the mapping of VAE to the implicit isometric embedding. Assume the data manifold is smooth and differentiable. Let $S_{\text{input}}(\subset \mathbb{R}^m)$ be an input space of the dataset. $D(\boldsymbol{x}, \acute{\boldsymbol{x}})$ denotes a metric for points $\boldsymbol{x}, \acute{\boldsymbol{x}} \in S_{\text{input}}$. Using the second order Taylor expansion, $D(\boldsymbol{x}, \boldsymbol{x} + \delta\boldsymbol{x})$ can be approximated by ${}^t\delta\boldsymbol{x} \, \boldsymbol{G_x}\delta\boldsymbol{x}$, where $\boldsymbol{G_x}$ and $\delta\boldsymbol{x}$ are an $\boldsymbol{x}$ dependent positive definite Hermitian metric tensor and an arbitrary infinitesimal displacement in $S_{\text{input}}$, respectively. The derivations of $\boldsymbol{G_x}$ for SSE, BCE, and SSIM are shown in Appendix H.2. Next, an implicit isometric embedding space $S_{\text{Iso}}(\subset \mathbb{R}^m)$ is introduced like the isometric latent space in RaDOGAGA (Kato et al., 2020), such that the entropy of data representation is minimum in the inner product space of $\boldsymbol{G_x}$. Let $\boldsymbol{y}$ and $y_j$ be a point in $S_{\text{Iso}}$ and its $j$-th component, respectively. Because of the isometricity, $p(\boldsymbol{x}) \simeq p(\boldsymbol{y})$ will hold. We will also show the posterior variance of each dimensional component $y_j$ is a constant $\beta/2$. In addition, the variance of $y_j$ will show the importance like PCA when the data manifold has a disentangled feature by nature in the metric space of $\boldsymbol{G_x}$ and the prior covariance is diagonal.

Then, $S_{\text{Iso}}$ is nonlinearly scaled to the VAE's anisometric orthogonal space $S_{\text{VAE}}(\subset \mathbb{R}^n)$ on a variable-by-variable basis. Let $\boldsymbol{z}$ be a point in $S_{\text{VAE}}$, and $z_j$ denotes the $j$-th component of $\boldsymbol{z}$. Let $p(y_j)$ and $p(z_j)$ be the probability distribution of the $j$-th variable in $S_{\text{Iso}}$ and $S_{\text{VAE}}$. Each variable $y_j$ is nonlinearly scaled to $z_j$, such that $\mathrm{d}z_j/\mathrm{d}y_j = p(y_j)/p(z_j)$ to fit the cumulative distribution. $\mathrm{d}z_j/\mathrm{d}y_j$ is $\sigma_{j(\boldsymbol{x})}/\sqrt{\beta/2}$, the ratio of posterior's standard deviations for $z_j$ and $y_j$, such that KL divergences in both spaces are equivalent. In addition, dimensional components whose KL divergences are zero can be discarded because such dimensions have no information.

## 3.2 Reformulation of objective to the form using $\partial \boldsymbol{x}/\partial z_j$ and $\partial \boldsymbol{x}/\partial \boldsymbol{z}$

We reformulate the objective $L_{\boldsymbol{x}}$ to the form using $\partial \boldsymbol{x}/\partial z_j$ and $\partial \boldsymbol{x}/\partial \boldsymbol{z}$. Here, the dimensions of $\boldsymbol{x}$ and $\boldsymbol{z}$, i.e., $m$ and $n$, are set as the same. The condition to reduce $n$ is shown in section 3.3.

**Reformulation of $D(\boldsymbol{x}, \hat{\boldsymbol{x}})$ loss:** In accordance with Kato et al. (2020), the loss $D(\boldsymbol{x}, \hat{\boldsymbol{x}})$ can be decomposed into $D(\check{\boldsymbol{x}}, \hat{\boldsymbol{x}}) + D(\boldsymbol{x}, \check{\boldsymbol{x}})$, where $\check{\boldsymbol{x}}$ denotes $\mathrm{Dec}_\theta(\boldsymbol{\mu}_{(\boldsymbol{x})})$. The first term $D(\check{\boldsymbol{x}}, \hat{\boldsymbol{x}})$ is a distortion between the decoded values of $\boldsymbol{\mu}_{(\boldsymbol{x})}$ with and without noise $\boldsymbol{\sigma}_{(\boldsymbol{x})}$. We call this term as a coding loss. This term is expanded as follows. $\delta \check{\boldsymbol{x}}$ denotes $\hat{\boldsymbol{x}} - \check{\boldsymbol{x}}$. Then, $D(\check{\boldsymbol{x}}, \hat{\boldsymbol{x}})$ term can be approximated by ${}^t\delta\check{\boldsymbol{x}}\, \boldsymbol{G}_{\boldsymbol{x}} \delta\check{\boldsymbol{x}}$. Let $\boldsymbol{x}_{z_j}$ be $\partial \boldsymbol{x}/\partial z_j$ at $z_j = \mu_{j(\boldsymbol{x})}$, and $\delta z_j \sim \mathcal{N}(0, \sigma_{j(\boldsymbol{x})})$ be an added noise in $z_j$. Then, $\delta\check{\boldsymbol{x}}$ is approximated by $\delta\check{\boldsymbol{x}} \simeq \sum_{j=1}^m \delta z_j\, \boldsymbol{x}_{z_j}$. Because $\delta z_j$ and $\delta z_k$ for $j \neq k$ are uncorrelated, the average of $D(\check{\boldsymbol{x}}, \hat{\boldsymbol{x}})$ over $\boldsymbol{z} \sim q_\phi(\boldsymbol{z}|\boldsymbol{x})$ can be finally reformulated by

$$E_{\boldsymbol{z}\sim q_\phi(\boldsymbol{z}|\boldsymbol{x})}\left[D(\check{\boldsymbol{x}}, \hat{\boldsymbol{x}})\right] \simeq E_{\boldsymbol{z}\sim q_\phi(\boldsymbol{z}|\boldsymbol{x})}\left[{}^t\delta\check{\boldsymbol{x}}\, \boldsymbol{G}_{\boldsymbol{x}} \delta\check{\boldsymbol{x}}\right] \simeq \sum_{j=1}^n \sigma_{j(\boldsymbol{x})}{}^2\, {}^t\boldsymbol{x}_{z_j} \boldsymbol{G}_x \boldsymbol{x}_{z_j}. \tag{5}$$

The second term $D(\boldsymbol{x}, \check{\boldsymbol{x}})$ is a loss between the input data and $\mathrm{Dec}_\theta(\boldsymbol{\mu}_{(\boldsymbol{x})})$. We call this term a transform loss. We presume VAE is analogous to the Wiener filter (Wiener, 1964; Jin et al., 2003) where the coding loss is regarded as an added noise. From the Wiener filter theory, the ratio between the transform loss and coding loss is close to the ratio between the coding loss and the variance of the input data. The coding loss, approximately $n\beta/2$ as in Eq. 14, should be smaller than the variance of the input data to capture meaningful information. Thus the transform loss, usually small, is not considered in the following discussion. Appendix B explains the detail in a simple 1-dimensional VAE. We show the exhaustive and quantitative evaluation of coding loss and transform loss in the toy dataset in appendix E.2 to validate this approximation.

**Reformulation of KL divergence:** When $\sigma_{j(\boldsymbol{x})} \ll 1$, $\sigma_{j(\boldsymbol{x})}{}^2 \ll -\log \sigma_{j(\boldsymbol{x})}{}^2$ is observed. For example, when $\sigma_{j(\boldsymbol{x})}{}^2 < 0.1$, we have $-(\sigma_{j(\boldsymbol{x})}{}^2/\log \sigma_{j(\boldsymbol{x})}{}^2) < 0.05$. In such dimensions, $D_{\mathrm{KL}j(x)}$ can be approximated as Eq. 6 by ignoring the $\sigma_{j(\boldsymbol{x})}{}^2$ term and setting $p(\mu_{j(\boldsymbol{x})})$ to $\mathcal{N}(z_j; 0, 1)$:

$$D_{\mathrm{KL}j(x)} \simeq \frac{1}{2}\left(\mu_{j(\boldsymbol{x})}{}^2 - \log \sigma_{j(\boldsymbol{x})}{}^2 - 1\right) = -\log\left(\sigma_{j(\boldsymbol{x})}\, p(\mu_{j(\boldsymbol{x})})\right) - \frac{\log 2\pi e}{2}. \tag{6}$$

Eq. 6 can be considered as a rate of entropy coding for a symbol with mean $\mu_{j(\boldsymbol{x})}$ allowing quantization noise $\sigma_{j(\boldsymbol{x})}{}^2$, as shown in Appendix H.3. Thus, in the dimension with meaningful information, $\sigma_{j(\boldsymbol{x})}{}^2$ is much smaller than the prior variance 1, and the approximation in Eq.6 is reasonable. Let $p(\boldsymbol{\mu}_{(\boldsymbol{x})})$ be $\prod_{j=1}^n p(\mu_{j(\boldsymbol{x})})$. $p(\boldsymbol{\mu}_{(\boldsymbol{x})}) = p(\boldsymbol{x})\,|\det(\partial \boldsymbol{x}/\partial \boldsymbol{z})|$ holds where $\det(\partial \boldsymbol{x}/\partial \boldsymbol{z})$ is a Jacobian determinant at $\boldsymbol{z} = \boldsymbol{\mu}_{(\boldsymbol{x})}$. Let $C_{D_{\mathrm{KL}}}$ be a constant $\frac{n}{2}\log 2\pi e$. Then, $D_{\mathrm{KL}}(\cdot)$ is reformulated by

$$D_{\mathrm{KL}}(\cdot) \simeq -\log\left(p(\boldsymbol{\mu}_{(\boldsymbol{x})})\prod_{j=1}^n \sigma_{j(\boldsymbol{x})}\right) - C_{D_{\mathrm{KL}}} \simeq -\log\left(p(\boldsymbol{x})\left|\det\left(\frac{\partial \boldsymbol{x}}{\partial \boldsymbol{z}}\right)\right|\prod_{j=1}^n \sigma_{j(\boldsymbol{x})}\right) - C_{D_{\mathrm{KL}}}. \tag{7}$$

**Final objective form:** From Eqs. 5 and 7, the objective $L'_{\boldsymbol{x}}$ to minimise is derived as:

$$L'_{\boldsymbol{x}} = \sum_{j=1}^n \sigma_{j(\boldsymbol{x})}{}^2\, {}^t\boldsymbol{x}_{z_j} \boldsymbol{G}_x \boldsymbol{x}_{z_j} - \beta \log\left(p(\boldsymbol{x})\left|\det\left(\frac{\partial \boldsymbol{x}}{\partial \boldsymbol{z}}\right)\right|\prod_{j=1}^n \sigma_{j(\boldsymbol{x})}\right) - C_{D_{\mathrm{KL}}}. \tag{8}$$

## 3.3 Proof of the hypothesis

**Mapping VAE to implicit isometric embedding:** The minimum condition of $L'_{\boldsymbol{x}}$ at $\boldsymbol{x}$ is examined. Let $\tilde{\boldsymbol{x}}_{z_j}$ be the $j$-th column vector of a cofactor matrix for Jacobian matrix $\partial \boldsymbol{x}/\partial \boldsymbol{z}$. Note that $\mathrm{d} \log |\det(\partial \boldsymbol{x}/\partial \boldsymbol{z})|/\mathrm{d}\boldsymbol{x}_{z_j} = \tilde{\boldsymbol{x}}_{z_j}/\det(\partial \boldsymbol{x}/\partial \boldsymbol{z})$ holds as is also used in Kato et al. (2020). Using this equation, the derivative of $L'_{\boldsymbol{x}}$ by $\boldsymbol{x}_{z_j}$ is described by

$$\frac{\mathrm{d}L'_{\boldsymbol{x}}}{\mathrm{d}\boldsymbol{x}_{z_j}} = 2\sigma_{j(\boldsymbol{x})}{}^2 \boldsymbol{G}_x \boldsymbol{x}_{z_j} - \frac{\beta}{\det(\partial \boldsymbol{x}/\partial \boldsymbol{z})}\tilde{\boldsymbol{x}}_{z_j}. \tag{9}$$

Note that ${}^t\boldsymbol{x}_{z_k} \cdot \tilde{\boldsymbol{x}}_{z_j} = \det(\partial \boldsymbol{x}/\partial \boldsymbol{z})\, \delta_{jk}$ holds by the cofactor's property. Here, $\cdot$ denotes the dot product, and $\delta_{jk}$ denotes the Kronecker delta. By setting Eq. 9 to zero and multiplying ${}^t\boldsymbol{x}_{z_k}$ from the left, the condition to minimize $L'_{\boldsymbol{x}}$ is derived by the next orthogonal form of $\boldsymbol{x}_{z_j}$:

$$(2\sigma_{j(\boldsymbol{x})}{}^2/\beta)\, {}^t\boldsymbol{x}_{z_k} \boldsymbol{G}_x \boldsymbol{x}_{z_j} = \delta_{jk}. \tag{10}$$

Here, the diagonal posterior covariance is the key for orthogonality. Next, implicit latent variable $\boldsymbol{y}$ and its $j$-th dimensional component $y_j$ are introduced. Set $y_j$ to zero at $z_j = 0$. The derivative between $y_j$ and $z_j$ at $\mu_{j(\boldsymbol{x})}$ is defined by

$$\frac{\mathrm{d}y_j}{\mathrm{d}z_j}\bigg|_{z_j=\mu_{j(\boldsymbol{x})}} = \sqrt{\frac{\beta}{2}}\sigma_{j(\boldsymbol{x})}^{-1}. \tag{11}$$

$\boldsymbol{x}_{y_j}$ denotes $\partial \boldsymbol{x}/\partial y_j$. By applying $\boldsymbol{x}_{z_j} = \mathrm{d}y_j/\mathrm{d}z_j\, \boldsymbol{x}_{y_j}$ to Eq. 10, $\boldsymbol{x}_{y_j}$ shows the isometric property (Han & Hong, 2006; Kato et al., 2020) in the inner product space with a metric tensor $\boldsymbol{G}_{\boldsymbol{x}}$ as follows:

$$^t\boldsymbol{x}_{y_j}\boldsymbol{G}_{\boldsymbol{x}}\boldsymbol{x}_{y_k} = \delta_{jk}. \tag{12}$$

**Minimum entropy of implicit isometric representation:** Let $L'_{\min \boldsymbol{x}}$ be a minimum of $L'_{\boldsymbol{x}}$ at $\boldsymbol{x}$. $D_{\min \boldsymbol{x}}$ and $R_{\min \boldsymbol{x}}$ denote a coding loss and KL divergence in $L'_{\min \boldsymbol{x}}$, respectively. By applying Eqs. 10-11 and $p(z_j) = (\mathrm{d}y_j/\mathrm{d}z_j)\,p(y_j)$ to Eqs. 5 and 7, the following equations are derived:

$$L'_{\min \boldsymbol{x}} = D_{\min \boldsymbol{x}} + \beta R_{\min \boldsymbol{x}}, \text{ where } D_{\min \boldsymbol{x}} = \frac{n\beta}{2}, \; R_{\min \boldsymbol{x}} = -\log p(\boldsymbol{y}) - \frac{n\log(\beta\pi e)}{2}. \tag{13}$$

Here, $D_{\min \boldsymbol{x}}$ is derived as $\sum_{j=1}^{n}(\beta/2)\,^t\boldsymbol{x}_{y_j}\boldsymbol{G}_{\boldsymbol{x}}\boldsymbol{x}_{y_j} = n\beta/2$, implying each dimensional posterior variance of the implicit isometric variable is a constant $\beta/2$. In addition, $\exp(-L'_{\min \boldsymbol{x}}/\beta) = p(\boldsymbol{y})\exp(\text{Const.}) \propto p(\boldsymbol{y}) \simeq p(\boldsymbol{x})$ will hold in the inner product space of $\boldsymbol{G}_{\boldsymbol{x}}$ from the isometricity.

By averaging $L'_{\min \boldsymbol{x}}$ over $\boldsymbol{x} \sim p(\boldsymbol{x})$ and approximating this average by the integration over $\boldsymbol{y} \sim p(\boldsymbol{y})$, the global minimum $L'_{\mathrm{G}}$ is derived as:

$$L'_{\mathrm{G}} = D_{\mathrm{G}} + \beta R_{\mathrm{G}}, \text{ where } D_{\mathrm{G}} = \frac{n\beta}{2}, \; R_{\mathrm{G}} = \min_{p(\boldsymbol{y})}\left(-\int p(\boldsymbol{y})\log p(\boldsymbol{y})\mathrm{d}\boldsymbol{y}\right) - \frac{n\log(\beta\pi e)}{2}. \tag{14}$$

The term $-\int p(\boldsymbol{y})\log p(\boldsymbol{y})\mathrm{d}\boldsymbol{y}$ in $R_{\mathrm{G}}$ is the entropy of $\boldsymbol{y}$. Thus, the optimal implicit isometric space is derived such that the entropy of data representation is minimum in the inner product space of $\boldsymbol{G}_{\boldsymbol{x}}$.

When the data manifold has a disentangled property in the given metric, each $y_j$ will capture a disentangled feature with minimum entropy, as shown in Kato et al. (2020). This is analogous to PCA for Gaussian data, which gives the disentangled representation with minimum entropy in SSE. Considering the similarity to the PCA eigenvalues, the variance of $y_j$ will indicate the importance of each dimension. In the dimensions where the variance of $y_j$ is less than $\beta/2$, $\sigma_{j(\boldsymbol{x})} = 1$, $\mu_{j(\boldsymbol{x})} = 0$, and $D_{\mathrm{KL}j(x)} = 0$ will hold. In addition, $\sigma_{j(\boldsymbol{x})}^2\,^t\boldsymbol{x}_{z_j}\boldsymbol{G}_{\boldsymbol{x}}\boldsymbol{x}_{z_j}$ will be close to 0 because this needs not to be balanced with $D_{\mathrm{KL}j(x)}$. This is similar to the case in the RD theory in Eq. 4 where $\sigma_{zj}^2$ is less than $d$, meaning no information. As a result, Eqs. 10-14 will not hold here. Thus, latent variables with variances from the largest to the n-th with $D_{\mathrm{KL}j(x)} > 0$ are sufficient for the representation and the dimensions with $D_{\mathrm{KL}j(x)} = 0$ can be ignored, allowing the reduction of the dimension $n$ for $\boldsymbol{z}$.

Some approximations may be slightly violated, however, our analysis still helps to understand VAE.

### 3.4 DISCUSSION AND RELATIONSHIP WITH PRIOR THEORETICAL STUDIES

First, we show $\beta$-VAE optimum as in Eq. 14 can be interpreted as the rate-distortion optimum (Eq. 4) in RD theory when the uniform distortion $d$ in Eq. 4 is set to $\beta/2$ in the metric defined space. $H(X) = -\int p(\boldsymbol{x})\log p(\boldsymbol{x})\,\mathrm{d}\boldsymbol{x}$ denotes a differential entropy for a set $\boldsymbol{x} \in X; \boldsymbol{x} \sim p(\boldsymbol{x})$. For the 1-dimensional Gaussian data $x \sim \mathcal{N}(x, 0, \sigma^2)$, $H(X) = \frac{1}{2}\log(2\pi e\sigma^2)$ holds. Thus, $R_{opt}$ in Eq. 4 is derived as a difference of the differential entropy between transformed data $\boldsymbol{z} \sim \prod_j \mathcal{N}(z_j; 0, \sigma_{z_j})$ and uniform distortion $\boldsymbol{D} \sim \mathcal{N}(\boldsymbol{D}; 0, d\boldsymbol{I}_m)$. $R_{\mathrm{G}}$ is also derived as a difference of the differential entropy between transformed data $\boldsymbol{y} \sim p(\boldsymbol{y})$ and uniform distortion $\boldsymbol{D} \sim \mathcal{N}(\boldsymbol{D}; 0, (\beta/2)\boldsymbol{I}_m)$. Furthermore, $D_{\mathrm{G}}$ in Eq. 14 can be interpreted as $D_{opt}$ in Eq. 4 by setting $d = \beta/2$. As a result, the VAE optimal corresponds to the rate-distortion optimal of transform coding in RD theory, and $\beta/2$ is regarded as a variance of the constant distortion equally added to each dimensional component. Because of the isometricity, the power of distortion (i.e., posterior variance) in the implicit isometric space is the same as that in the metric defined input space. Thus the conditional distribution after optimization in the metric defined space is derived as $p_\theta(\boldsymbol{x}|\boldsymbol{z}) = p_\theta(\boldsymbol{x}|\hat{\boldsymbol{x}}) \simeq \mathcal{N}(\boldsymbol{x}; \hat{\boldsymbol{x}}, (\beta/2)\boldsymbol{I})$. This is consistent with the fact that the quality of the reconstructed data becomes worse in larger $\beta$.

Next, we estimate the reconstruction loss $E_{q_\phi(\boldsymbol{z}|\boldsymbol{x})}[\log p_\theta(\boldsymbol{x}|\boldsymbol{z})]$ and KL divergence $D_{\mathrm{KL}}(\cdot)$ in $\beta$-VAE and also correct the analysis in Alemi et al. (2018). Let $H = -E_{p(\boldsymbol{x})}[\log p(\boldsymbol{x})]$ be a differential entropy of input data. When $\beta = 1$, Alemi et al. (2018) suggest "the ELBO objective alone (and the marginal likelihood) cannot distinguish between models that make no use of the latent variable (autodecoders) versus models that make large use of the latent variable and learn useful representations for reconstruction (autoencoders)," because the reconstruction loss and KL divergence can be arbitrary value on the line $-E_{q_\phi(\boldsymbol{z}|\boldsymbol{x})}[\log p_\theta(\boldsymbol{x}|\boldsymbol{z})] + D_{\mathrm{KL}}(\cdot) = H$. Correctly, the reconstruction loss and KL divergence after optimization are deterministically estimated at any $\beta$ (including $\beta = 1$) as:

$$E_{q_\phi(\boldsymbol{z}|\boldsymbol{x})}[\log p_\theta(\boldsymbol{x}|\boldsymbol{z})] \simeq -(n/2)\log(\beta\pi e), \; D_{\mathrm{KL}}(\cdot) \simeq -\log p(\boldsymbol{y}) - (n/2)\log(\beta\pi e). \quad (15)$$

The proof is explained in Appendix A.1. Thus ELBO can be estimated as:

$$\mathrm{ELBO} = E_{p(\boldsymbol{x})}[E_{\boldsymbol{z} \sim q_\phi(\boldsymbol{z}|\boldsymbol{x})}[\log p_\theta(\boldsymbol{x}|\boldsymbol{z})] - D_{\mathrm{KL}}(\cdot)] \simeq E_{p(\boldsymbol{x})}[\log p(\boldsymbol{y})] \simeq E_{p(\boldsymbol{x})}[\log p(\boldsymbol{x})]. \quad (16)$$

As a result, when the objective of $\beta$-VAE is optimised, ELBO (Eq. 1) in the original form (Kingma & Welling, 2014) is approximately equal to the log-likelihood of $\boldsymbol{x}$, regardless $\beta = 1$ or not.

Finally, the predetermined conditional distribution $p_{\mathbb{R}p}(\boldsymbol{x}|\hat{\boldsymbol{x}})$ and the true conditional distribution after optimization $p_{\mathbb{R}\theta}(\boldsymbol{x}|\hat{\boldsymbol{x}})$ are examined using $\beta$ in the input Euclidean space of $\boldsymbol{x}$. Assume $p_{\mathbb{R}p}(\boldsymbol{x}|\hat{\boldsymbol{x}}) = \mathcal{N}(\boldsymbol{x}; \hat{\boldsymbol{x}}, \sigma^2 \boldsymbol{I})$. In this case, the metric $D(\boldsymbol{x}, \hat{\boldsymbol{x}})$ is derived as $-\log p_{\mathbb{R}p}(\boldsymbol{x}|\hat{\boldsymbol{x}}) = (1/2\sigma^2)|\boldsymbol{x} - \hat{\boldsymbol{x}}|_2^2 + \mathrm{Const}$. From Eq. 13, the following equations are derived:

$$E_{q_\phi(\hat{\boldsymbol{x}}|\boldsymbol{x})}[D(\boldsymbol{x}, \hat{\boldsymbol{x}})] = E_{q_\phi(\hat{\boldsymbol{x}}|\boldsymbol{x})}\left[\frac{1}{2\sigma^2}|\boldsymbol{x} - \hat{\boldsymbol{x}}|_2^2\right] = E_{q_\phi(\hat{\boldsymbol{x}}|\boldsymbol{x})}\left[\frac{1}{2\sigma^2}\sum_i (x_i - \hat{x}_i)^2\right] \simeq n\beta/2, \quad (17)$$

$$E_{q_\phi(\hat{\boldsymbol{x}}|\boldsymbol{x})}\left[(x_i - \hat{x}_i)^2\right] \simeq \beta\sigma^2. \quad (18)$$

Because the variance of each dimension is estimated as $\beta\sigma^2$, the true conditional distribution after optimization is approximated as $p_{\mathbb{R}\theta}(\boldsymbol{x}|\hat{\boldsymbol{x}}) \simeq \mathcal{N}(\boldsymbol{x}; \hat{\boldsymbol{x}}, \beta\sigma^2 \boldsymbol{I})$. If $\beta = 1$, i.e., the original VAE, $p_{\mathbb{R}p}(\boldsymbol{x}|\hat{\boldsymbol{x}})$ and $p_{\mathbb{R}\theta}(\boldsymbol{x}|\hat{\boldsymbol{x}})$ are equivalent as expected. If $\beta \neq 1$, however, $p_{\mathbb{R}p}(\boldsymbol{x}|\hat{\boldsymbol{x}})$ and $p_{\mathbb{R}\theta}(\boldsymbol{x}|\hat{\boldsymbol{x}})$ are different. Actually, what $\beta$-VAE does is only to scale the variance of the pre-determined conditional distribution in the original VAE by a factor of $\beta$, because $\beta$-VAE objective can be rewritten as:

$$E_{q_\phi(\cdot)}[\log \mathcal{N}(\boldsymbol{x}; \hat{\boldsymbol{x}}, \sigma^2 \boldsymbol{I})] - \beta D_{\mathrm{KL}}(\cdot) = \beta\left(E_{q_\phi(\cdot)}[\log \mathcal{N}(\boldsymbol{x}; \hat{\boldsymbol{x}}, \beta\sigma^2 \boldsymbol{I})] - D_{\mathrm{KL}}(\cdot)\right) + \mathrm{const}. \quad (19)$$

More detailed discussions about prior works (Higgins et al. (2017); Alemi et al. (2018); Dai et al. (2018); Dai & Wipf (2019); Tishby et al. (1999); Goyal (2001)) are explained in Appendix A.

## 3.5 QUANTITATIVE PROPERTIES TO VALIDATE THE THEORY

This section shows three quantitative properties in VAE with a prior $\mathcal{N}(\boldsymbol{z}; 0, \boldsymbol{I}_n)$, to validate the theory in section 3.3. The second and third properties also provide practical data analysis approaches. The derivation of equations in the second and third properties are explained in appendix C.

**Norm of $\boldsymbol{x}_{y_j}$ equal to 1:** Let $\boldsymbol{e}^{(j)}$ be a vector $(0, \cdots, \overset{j\text{-th}}{1}, \cdots, 0)$ where the $j$-th dimension is 1, and others are 0. Let $D'_j(\boldsymbol{z})$ be $D(\mathrm{Dec}_\theta(\boldsymbol{z}), \mathrm{Dec}_\theta(\boldsymbol{z} + \epsilon\boldsymbol{e}^{(j)}))/\epsilon^2$, where $\epsilon$ denotes a minute value for the numerical differential. From Eq. 10, the squared norm of $\boldsymbol{x}_{y_j}$ can be numerically evaluated as the first term of Eq. 20. This value will be equal to 1 at any $\boldsymbol{x}$ and dimension $j$ except $D_{\mathrm{KL}j(x)} = 0$.

$$\frac{2}{\beta}\sigma_{j(\boldsymbol{x})}^2 D'_j(\boldsymbol{z}) \simeq \frac{2}{\beta}\left(\sigma_{j(\boldsymbol{x})}^2 \, {}^t\boldsymbol{x}_{z_j}\boldsymbol{G}_x\boldsymbol{x}_{z_j}\right) \simeq {}^t\boldsymbol{x}_{y_j}\boldsymbol{G}_x\boldsymbol{x}_{y_j} = 1. \quad (20)$$

If observed, the existence of an implicit isometric embedding can be shown because of unit norm and orthogonality (Rolínek et al., 2019). Eq. 20 also show $\sigma_{j(\boldsymbol{x})}^2 \, {}^t\boldsymbol{x}_{z_j}\boldsymbol{G}_x\boldsymbol{x}_{z_j} \simeq \frac{\beta}{2}$, implying that a noise $\sigma_{j(\boldsymbol{x})}$ added to each dimension of latent variable causes an equal noise $\beta/2$ in the input space.

**PCA-like feature:** When the data manifold has a disentangled property in the given metric, the variance of the $j$-th implicit latent component $y_j$ can be roughly estimated as

$$\int y_j^2 p(y_j)\mathrm{d}y_j \simeq \frac{\beta}{2} \underset{\boldsymbol{x} \sim p(\boldsymbol{x})}{E}[\sigma_{j(\boldsymbol{x})}^{-2}]. \quad (21)$$

The average $E[\sigma_{j(\boldsymbol{x})}^{-2}]$ on the right allows evaluating the quantitative importance of each dimension in practice, like the eigenvalue of PCA. Note that a dimension whose average is close to 1 implies $D_{\mathrm{KL}j(x)} = 0$. Such a dimension has no information and is an exceptions of the property in Eq. 20.

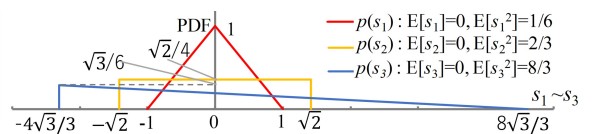

Figure 2: PDFs of three variables to generate a toy dataset.

**Estimation of the data probability distribution:** First, assume the case $m = n$. Since the $\boldsymbol{y}$ space is isometric to the inner product space of $\boldsymbol{G}_x$, the PDFs in both spaces are the same. The Jacobian determinant between the input space and inner product space, giving the the ratio of PDFs, is derived as $|\boldsymbol{G}_x|^{\frac{1}{2}}$. We set $p(\boldsymbol{\mu}_{(\boldsymbol{x})})$ to the prior. Thus, the data probability in the input space can be estimated by $|\boldsymbol{G}_x|^{\frac{1}{2}}$ and either the prior/posterior or $L_{\boldsymbol{x}}$ after training, as the following last two equations:

$$p(\boldsymbol{x}) \simeq |\boldsymbol{G}_x|^{\frac{1}{2}} p(\boldsymbol{y}) \quad \propto |\boldsymbol{G}_x|^{\frac{1}{2}} \, p(\boldsymbol{\mu}_{(\boldsymbol{x})}) \prod_{j=1}^{m} \sigma_{j(\boldsymbol{x})} \quad \propto |\boldsymbol{G}_x|^{\frac{1}{2}} \exp\left(-\frac{1}{\beta} L_{\boldsymbol{x}}\right). \tag{22}$$

In the case $m > n$, the derivation of the PDF ratio between the input space and the inner product space is generally intractable, except for $\boldsymbol{G}_x = a_{\boldsymbol{x}} \boldsymbol{I}_m$, where $a_{\boldsymbol{x}}$ is an $\boldsymbol{x}$-dependent scalar factor. In this case, the PDF ratio is given by $a_x{}^{n/2}$. Thus, $p(\boldsymbol{x})$ can be estimated as follows:

$$p(\boldsymbol{x}) \quad \propto a_{\boldsymbol{x}}^{\frac{n}{2}} \, p(\boldsymbol{\mu}_{(\boldsymbol{x})}) \prod_{j=1}^{n} \sigma_{j(\boldsymbol{x})} \quad \propto a_{\boldsymbol{x}}^{\frac{n}{2}} \exp\left(-\frac{1}{\beta} L_{\boldsymbol{x}}\right). \tag{23}$$

Equations 22 and 23 enable a probability-based quantitative data analysis/sampling in practice.

## 4 EXPERIMENT

We show the experiments of the quantitative properties presented in Section 3.5. First, the results of the toy dataset are presented. Then, the results of CelebA are shown as a real data example.

### 4.1 EVALUATION OF QUANTITATIVE PROPERTIES IN THE TOY DATASET

The toy dataset is generated as follows. First, three dimensional variables $s_1$, $s_2$, and $s_3$ are sampled in accordance with the three different shapes of distributions $p(s_1)$, $p(s_2)$, and $p(s_3)$, as shown in Fig. 2. The variances of $s_1$, $s_2$, and $s_3$ are $1/6$, $2/3$, and $8/3$, respectively, such that the ratio of the variances is 1:4:16. Second, three 16-dimensional uncorrelated vectors $\boldsymbol{v}_1$, $\boldsymbol{v}_2$, and $\boldsymbol{v}_3$ with L2 norm 1, are provided. Finally, $50,000$ toy data with 16 dimensions are generated by $\boldsymbol{x} = \sum_{i=1}^{3} s_i \boldsymbol{v}_i$. The data generation probability $p(\boldsymbol{x})$ is also set to $p(s_1)p(s_2)p(s_3)$. If our hypothesis is correct, $p(y_j)$ will be close to $p(s_j)$. Then, $\sigma_{j(\boldsymbol{x})} \propto \mathrm{d}z_j/\mathrm{d}y_j = p(y_j)/p(z_j)$ will also vary a lot with these varieties of PDFs. Because the properties presented in Section 3.5 are calculated from $\sigma_{j(\boldsymbol{x})}$, our theory can be easily validated by evaluating those properties.

Then, the VAE model is trained using Eq. 1. We use two kinds of the reconstruction loss $D(\cdot, \cdot)$ to analyze the effect of the loss metrics. The first is the square error loss equivalent to sum square error (SSE). The second is the downward-convex loss which we design as Eq. 24, such that the shape becomes similar to the BCE loss as in Appendix H.2:

$$D(\boldsymbol{x}, \hat{\boldsymbol{x}}) = a_{\boldsymbol{x}} \|\boldsymbol{x} - \hat{\boldsymbol{x}}\|_2^2, \quad \text{where } a_{\boldsymbol{x}} = (2/3 + 2 \|\boldsymbol{x}\|_2^2/21) \text{ and } \boldsymbol{G}_x = a_{\boldsymbol{x}} \boldsymbol{I}_m. \tag{24}$$

Here, $a_{\boldsymbol{x}}$ is chosen such that the mean of $a_{\boldsymbol{x}}$ for the toy dataset is 1.0 since the variance of $\boldsymbol{x}$ is 1/6+2/3+8/3=7/2. The details of the networks and training conditions are written in Appendix D.1.

Then the network is trained with two types of reconstruction losses. The ratio of transform loss to coding loss for the square error loss is 0.023, and that for the downward-convex loss is 0.024. As expected in section 3.2, the transform losses are negligibly small. Tables 1 and 2 show the measurements of $\frac{2}{\beta} \sigma_{j(\boldsymbol{x})}{}^2 D'_j(\boldsymbol{z})$ (shown as $\frac{2}{\beta} \sigma_j{}^2 D'_j$), $D'_j(\boldsymbol{z})$, and $\sigma_{j(\boldsymbol{x})}{}^{-2}$ described in Section 3.5. In these tables, $z_1$, $z_2$, and $z_3$ show acquired latent variables. "Av." and "SD" are the average and standard deviation, respectively. To begin with, the norm of the implicit orthonormal basis

Table 1: Property measurements of the toy dataset trained with the square error loss.

| variable | | $z_1$ | $z_2$ | $z_3$ |
|---|---|---|---|---|
| $\frac{2}{\beta}\sigma_j{}^2 D'_j$ | Av. | 0.965 | 0.925 | 0.972 |
| | SD | 0.054 | 0.164 | 0.098 |
| $D'_j(\boldsymbol{z})$ | Av. | 0.162 | 0.726 | 2.922 |
| | SD | 0.040 | 0.466 | 1.738 |
| $\sigma_{j(\boldsymbol{x})}{}^{-2}$ | Av. | 3.33e1 | 1.46e2 | 5.89e2 |
| (Ratio) | Av. | 1.000 | 4.39 | 17.69 |

Table 2: Property measurements of the toy dataset trained with the downward-convex loss.

| variable | | $z_1$ | $z_2$ | $z_3$ |
|---|---|---|---|---|
| $\frac{2}{\beta}\sigma_j{}^2 D'_j$ | Av. | 0.964 | 0.928 | 0.978 |
| | SD | 0.060 | 0.160 | 0.088 |
| $D'_j(\boldsymbol{z})$ | Av. | 0.161 | 0.696 | 2.695 |
| | SD | 0.063 | 0.483 | 1.573 |
| $\sigma_{j(\boldsymbol{x})}{}^{-2}$ | Av. | 3.30e1 | 1.40e2 | 5.43e2 |
| (Ratio) | Av. | 1.000 | 4.25 | 16.22 |

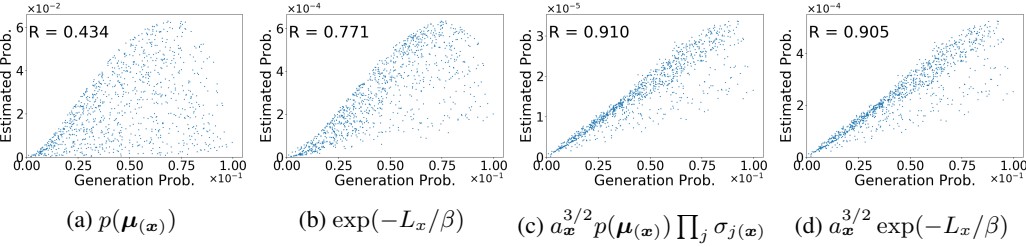

(a) $p(\boldsymbol{\mu}_{(\boldsymbol{x})})$    (b) $\exp(-L_x/\beta)$    (c) $a_{\boldsymbol{x}}^{3/2} p(\boldsymbol{\mu}_{(\boldsymbol{x})}) \prod_j \sigma_{j(\boldsymbol{x})}$    (d) $a_{\boldsymbol{x}}^{3/2} \exp(-L_x/\beta)$

Figure 3: Scattering plots of the data generation probability (x-axis) versus four estimated probabilities (y-axes) for the downward-convex loss. y-axes are (a) $p(\boldsymbol{\mu}_{(\boldsymbol{x})})$, (b) $\exp(-L_x/\beta)$, (c) $a_{\boldsymbol{x}}^{3/2} p(\boldsymbol{\mu}_{(\boldsymbol{x})}) \prod_j \sigma_{j(\boldsymbol{x})}$, and (d) $a_{\boldsymbol{x}}^{3/2} \exp(-L_x/\beta)$.

is discussed. In both tables, the values of $\frac{2}{\beta}\sigma_{(\boldsymbol{x})j}{}^2 D'_j(\boldsymbol{z})$ are close to 1.0 in each dimension as described in Eq. 23. By contrast, the average of $D'_j(\boldsymbol{z})$, which corresponds to ${}^t\boldsymbol{x}_{z_j} \boldsymbol{G}_x \boldsymbol{x}_{z_k}$, is different in each dimension. Therefore, the derivative of $\boldsymbol{x}$ with $\boldsymbol{z}_j$, the original latent variable of VAE, is not normalized.

Next, the PCA-like feature is examined. The average of $\sigma_{j(\boldsymbol{x})}{}^{-2}$ in Eq.21 and its ratio are shown in Tables 1 and 2. Although the average of $\sigma_{j(\boldsymbol{x})}{}^{-2}$ is a rough estimation of variance, the ratio is close to 1:4:16, i.e., the variance ratio of generation parameters $s_1$, $s_2$, and $s_3$. When comparing both losses, the ratio of $s_2$ and $s_3$ for the downward-convex loss is somewhat smaller than that for the square error. This is explained as follows. In the downward-convex loss, $|\boldsymbol{x}_{y_j}|^2$ tends to be $1/a_{\boldsymbol{x}}$ from Eq. 12, i.e. ${}^t\boldsymbol{x}_{y_j} (a_x \boldsymbol{I}_m) \boldsymbol{x}_{y_k} = \delta_{jk}$. Therefore, the region in the inner product space with a larger norm is shrunk, and the estimated variances corresponding to $s_2$ and $s_3$ become smaller.

Figure 3 shows the scattering plots of the data generation probability $p(\boldsymbol{x})$ and estimated probabilities for the downward-convex loss. The plots for the square error loss are shown in Appendix E. Figure 3a shows the plots of $p(\boldsymbol{x})$ and the prior probabilities $p(\boldsymbol{\mu}_{(\boldsymbol{x})})$. This graph implies that it is difficult to estimate $p(\boldsymbol{x})$ only from the prior. The correlation coefficient shown as "R" (0.434) is also low. Figure 3b shows the plots of $p(\boldsymbol{x})$ and $\exp(-L_{\boldsymbol{x}}/\beta)$, i.e., the lower bound of likelihood. The correlation coefficient (0.771) becomes better, but is still not high. Next, Figures 3c and 3d show the plots of $a_{\boldsymbol{x}}^{3/2} p(\boldsymbol{\mu}_{(\boldsymbol{x})}) \prod_j \sigma_{j(\boldsymbol{x})}$ and $a_{\boldsymbol{x}}^{3/2} \exp(-L_{\boldsymbol{x}}/\beta)$ in Eq. 23. These graphs, showing a high correlation coefficients around 0.91, support that the objective $L_{\boldsymbol{x}}$ in Eq. 3 is optimized in the inner product space of $\boldsymbol{G}_x$. In the case of the square error loss, the plots with $\exp(-L_{\boldsymbol{x}}/\beta)$ also shows a high correlation coefficient 0.904 because $a_{\boldsymbol{x}}$ is 1, allowing the probability estimation from $L_{\boldsymbol{x}}$ in Eq. 3. The ablation study with different PDF, losses, and $\beta$ is shown in Appendix E.

## 4.2 EVALUATIONS IN CELEBA DATASET

This section evaluates the first and second quantitative properties of VAE trained with the CelebA dataset [1] (Liu et al., 2015) as an example of real data. This dataset is composed of 202,599 celebrity images. In use, the images are center-cropped to form $64 \times 64$ sized images.

---

[1](http://mmlab.ie.cuhk.edu.hk/projects/CelebA.html)

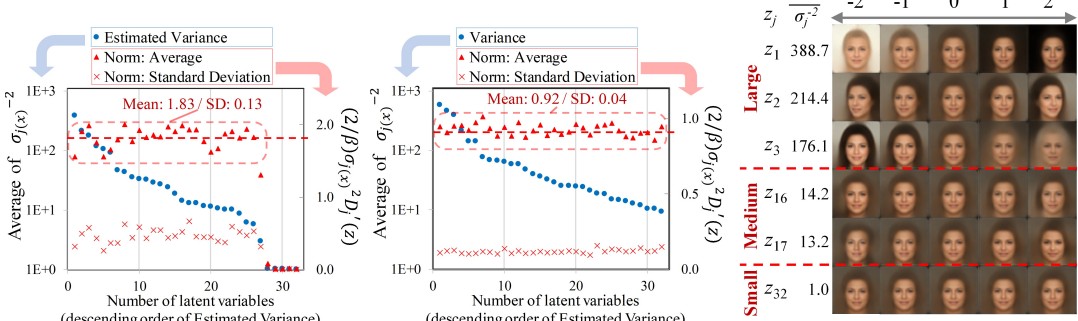

Figure 4: Graph of $\sigma_{j(\boldsymbol{x})}^{-2}$ average and $\frac{2}{\beta}\sigma_{j(\boldsymbol{x})}{}^2 D_j'(\boldsymbol{z})$ in VAE for CelebA dataset.

Figure 5: Graph of $\sigma_{j(\boldsymbol{x})}^{-2}$ average and $\frac{2}{\beta}\sigma_{j(\boldsymbol{x})}{}^2 D_j'(\boldsymbol{z})$ in VAE for CelebA dataset with explicit decomposed loss.

Figure 6: Dependency of decoded image changes with $z_j = -2$ to 2 on the average of $\sigma_{j(\boldsymbol{x})}^{-2}$.

We use SSIM, which is popular in image compression, as a reconstruction loss. The details of networks and training conditions are written in Appendix D.2.

Figure 4 shows the averages of $\sigma_{j(\boldsymbol{x})}^{-2}$ in Eq.21 as the estimated variances, as well as the average and the standard deviation of $\frac{2}{\beta}\sigma_{j(\boldsymbol{x})}{}^2 D_j'(\boldsymbol{z})$ in Eq.20 as the estimated square norm of implicit transform. The latent variables $z_i$ are numbered in descending order by the estimated variance. In the dimensions greater than the 27th, the averages of $\sigma_{j(\boldsymbol{x})}^{-2}$ are close to 1 and that of $\frac{2}{\beta}\sigma_{j(\boldsymbol{x})}{}^2 D_j'(\boldsymbol{z})$ is close to 0, implying $D_{\mathrm{KL}}(\cdot) = 0$. Between the 1st and 26th dimensions, the mean and standard deviation of $\frac{2}{\beta}\sigma_{j(\boldsymbol{x})}{}^2 D_j'(\boldsymbol{z})$ averages are 1.83 and 0.13, respectively. These values seem almost constant with a small standard deviation; however, the mean is somewhat larger than the expected value 1. This result implies that the implicit transform can be considered as almost orthonormal by dividing $\sqrt{1.83} \simeq 1.35$. Thus, the average of $\sigma_{j(\boldsymbol{x})}^{-2}$ still can determine the quantitative importance of each latent variable. This also mean that the added noise to each $y_j$ is around $1.83(\beta/2)$. We also train VAE by the decomposed loss explicitly, where $L_{\boldsymbol{x}}$ is set to $D(\boldsymbol{x}, \breve{\boldsymbol{x}}) + D(\breve{\boldsymbol{x}}, \hat{\boldsymbol{x}}) + \beta D_{\mathrm{KL}}(\cdot)$. Figure 5 shows the result. Here, the mean and standard deviation of $\frac{2}{\beta}\sigma_{j(\boldsymbol{x})}{}^2 D_j'(\boldsymbol{z})$ averages are 0.92 and 0.04, respectively, which suggests almost a unit norm. As a result, the explicit use of decomposed loss matches the theory better, allowing better analysis. The slight violation of the norm in the conventional form needs a more exact analysis as a future study.

Figure 6 shows decoder outputs where the selected latent variables are traversed from $-2$ to 2 while setting the rest to 0. The average of $\sigma_{j(\boldsymbol{x})}^{-2}$ is also shown there. The components are grouped by the average of $\sigma_{j(\boldsymbol{x})}^{-2}$, such that $z_1, z_2, z_3$ to the large, $z_{16}, z_{17}$ to the medium, and $z_{32}$ to the small, accordingly. In the large group, significant changes of background brightness, the direction of the face, and hair color are observed. In the medium group, we can see minor changes such as facial expressions. However, in the small group, there are almost no changes. This result strongly supports that the average of $\sigma_{j(\boldsymbol{x})}^{-2}$ shows the importance of each latent variable. The traversed outputs for all the component and results with another conditions are shown in Appendix F.

## 5 CONCLUSION

This paper provides a quantitative understanding of VAE by non-linear mapping to an isometric embedding. According to the Rate-distortion theory, the optimal transform coding is achieved by using PCA/KLT orthonormal transform, where the transform space is isometric to the input. From this analogy, we show theoretically and experimentally that VAE can be mapped to an implicit isometric embedding with a scale factor derived from the posterior parameter. Based on this property, we also clarify that VAE can provide a practical quantitative analysis of input data such as the probability estimation in the input space and the PCA-like quantitative multivariate analysis. We believe the quantitative properties thoroughly uncovered in this paper will be a milestone to further advance the information theory-based generative models such as VAE in the right direction.

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

## A  DETAILED RELATION TO PRIOR WORKS

Firstly, we clarify the the difference between ELBO in Eq. 1 and the objective $L_{\boldsymbol{x}}$ in Eq. 3 in the right direction. Then, we discuss the relation to the prior works. We also point out the incorrectness of some works.

### A.1  DERIVATION OF ELBO WITH CLEAR AND QUANTITATIVE FORM

We derive the reconstruction loss and KL divergence terms in ELBO (without $\beta$) at $\boldsymbol{x}$ in Eq. 1 when the objective of $\beta$-VAE $L'_{\boldsymbol{x}}$ in Eq. 13 is optimised. The reconstruction loss can be rewritten as

$$E_{\boldsymbol{z} \sim q_\phi(\boldsymbol{z}|\boldsymbol{x})}[\log p_\theta(\boldsymbol{x}|\boldsymbol{z})] = \int q_\phi(\boldsymbol{z}|\boldsymbol{x}) \log p_\theta(\boldsymbol{x}|\boldsymbol{z}) \mathrm{d}\boldsymbol{z} = \int q_\phi(\boldsymbol{y}|\boldsymbol{x}) \log p_\theta(\boldsymbol{x}|\boldsymbol{y}) \mathrm{d}\boldsymbol{y}. \quad (25)$$

Let $\boldsymbol{\mu}_{\boldsymbol{y}(\boldsymbol{x})}$ be a implicit isometric variable corresponding to $\mu_{(\boldsymbol{x})}$. Because the posterior variance in each isometric latent variable is a constant $\beta/2$, $q_\phi(\boldsymbol{y}|\boldsymbol{x}) \simeq \mathcal{N}(\boldsymbol{y}; \boldsymbol{\mu}_{\boldsymbol{y}(\boldsymbol{x})}, (\beta/2)\boldsymbol{I}_n)$ will hold. If $\beta/2$ is small, $p(\hat{\boldsymbol{x}}) \simeq p(\boldsymbol{x})$ will hold. Then, the next equation will hold also using isometricity;

$$p_\theta(\boldsymbol{x}|\boldsymbol{z}) = p_\theta(\boldsymbol{x}|\boldsymbol{y}) = p_\theta(\boldsymbol{x}|\hat{\boldsymbol{x}}) = p(\hat{\boldsymbol{x}}|\boldsymbol{x})p(\boldsymbol{x})/p(\hat{\boldsymbol{x}}) \simeq p(\hat{\boldsymbol{x}}|\boldsymbol{x}) \simeq q_\phi(\boldsymbol{y}|\boldsymbol{x}). \quad (26)$$

Thus the reconstruction loss is estimated as:

$$\begin{aligned} E_{\boldsymbol{z} \sim q_\phi(\boldsymbol{z}|\boldsymbol{x})}[\log p_\theta(\boldsymbol{x}|\boldsymbol{z})] \quad &\sim \quad \int \mathcal{N}(\boldsymbol{y}; \boldsymbol{\mu}_{\boldsymbol{y}(\boldsymbol{x})}, (\beta/2)\boldsymbol{I}_n) \log \mathcal{N}(\boldsymbol{y}; \boldsymbol{\mu}_{\boldsymbol{y}(\boldsymbol{x})}, (\beta/2)\boldsymbol{I}_n) \, \mathrm{d}\boldsymbol{y} \\ &= \quad -(n/2)\log(\beta\pi e). \end{aligned} \quad (27)$$

From Eq. 13, KL divergence is derived as:

$$D_{\mathrm{KL}}(\cdot) = R_{\min\boldsymbol{x}} = -\log p(\boldsymbol{y}) - (n/2)\log(\beta\pi e). \quad (28)$$

By summing both terms, ELBO at $\boldsymbol{x}$ can be estimated as

$$\begin{aligned} ELBO \quad &= \quad E_{\boldsymbol{x} \sim p(\boldsymbol{x})}[E_{\boldsymbol{z} \sim q_\phi(\boldsymbol{z}|\boldsymbol{x})}[\log p_\theta(\boldsymbol{x}|\boldsymbol{z})] - D_{\mathrm{KL}}(\cdot)] \\ &\simeq \quad E_{\boldsymbol{x} \sim p(\boldsymbol{x})}[\log p(\boldsymbol{y})] \\ &\simeq \quad E_{\boldsymbol{x} \sim p(\boldsymbol{x})}[\log p(\boldsymbol{x})]. \end{aligned} \quad (29)$$

As a result, ELBO (Eq. 1) in the original form (Kingma & Welling, 2014) is close to the log-likelihood of $\boldsymbol{x}$, regardless $\beta = 1$ or not, when the objective of $\beta$-VAE (Higgins et al., 2017) is optimised.

Some of the prior VAE works do not explicitly distinguish between the reconstruction loss $E_{\hat{\boldsymbol{x}} \sim p(\hat{\boldsymbol{x}}|\boldsymbol{x})}[p(\boldsymbol{x}|\hat{\boldsymbol{x}})]$ in ELBO and the distortion $D(\boldsymbol{x}, \hat{\boldsymbol{x}})$ in the objective $L_{\boldsymbol{x}}$ by mistake, which leads to some incorrect discussion.

In addition, there have also been incorrect discussions in some prior works. In ELBO derivation, they use $E_{\hat{\boldsymbol{x}} \sim p(\hat{\boldsymbol{x}}|\boldsymbol{x})}[p(\boldsymbol{x}|\hat{\boldsymbol{x}})]$ as a reconstruction loss, without discussing what kinds of properties the distortion probability $p(\hat{\boldsymbol{x}}|\boldsymbol{x})$ should be. In training VAE with a real dataset, by contrast, they use a predetermined distortion metric $D(\boldsymbol{x}, \hat{\boldsymbol{x}})$ like BCE and SSE as a reconstruction loss instead of a log-likelihood of the distortion probability, without discussing what distortion probability should be after optimization.

Correctly, the distortion probability $p(\hat{\boldsymbol{x}}|\boldsymbol{x})$ after training is determined by $\beta$ and the metric as $p_\theta(\hat{\boldsymbol{x}}|\boldsymbol{x}) \simeq \mathcal{N}(\hat{\boldsymbol{x}}; \boldsymbol{x}, (\beta/2)\boldsymbol{I}_m)$ in the metric defined space. Then, by applying $p_\theta(\hat{\boldsymbol{x}}|\boldsymbol{x}) \simeq p_\theta(\hat{\boldsymbol{x}}|\boldsymbol{x})$ to Eq. 1, the value of ELBO will become $\log p(\boldsymbol{y}) \simeq \log p(\boldsymbol{x})$ regardless $\beta = 1$ or not.

If $D(\boldsymbol{x}, \boldsymbol{x}+\delta\boldsymbol{x}) = {}^t\delta\boldsymbol{x}\boldsymbol{G}_{\boldsymbol{x}}\delta\boldsymbol{x} + O(||\delta\boldsymbol{x}||^3)$ is not SSE, by introducing a variable $\acute{\boldsymbol{x}} = \boldsymbol{L}_{\boldsymbol{x}}^{-1}\boldsymbol{x}$ where $\boldsymbol{L}_{\boldsymbol{x}}$ satisfies ${}^t\boldsymbol{L}_{\boldsymbol{x}}\boldsymbol{L}_{\boldsymbol{x}} = \boldsymbol{G}_{\boldsymbol{x}}$, the metric $D(\cdot, \cdot)$ can be replaced by SSE in the Euclidean space of $\acute{\boldsymbol{x}}$.

### A.2  RELATION TO TISHBY ET AL. (1999)

The theory described in Tishby et al. (1999) is consistent with our analysis. Tishby et al. (1999) clarified the behaviour of the compressed representation when the rate-distortion trade-off is optimized.

$\boldsymbol{x} \in X$ denotes the signal space with a fixed probability $p(\boldsymbol{x})$ and $\hat{\boldsymbol{x}} \in \hat{X}$ denotes its compressed representation. Let $D(\boldsymbol{x}, \hat{\boldsymbol{x}})$ be a loss metric. Then the rate-distortion trade-off can be described as:

$$L = I(X; \hat{X}) + \beta' \underset{p(\boldsymbol{x}, \hat{\boldsymbol{x}})}{E} [D(\boldsymbol{x}, \hat{\boldsymbol{x}})]. \tag{30}$$

By solving this condition, they derive the following equation:

$$p(\hat{\boldsymbol{x}}|\boldsymbol{x}) \propto \exp(-\beta' D(\boldsymbol{x}, \hat{\boldsymbol{x}})). \tag{31}$$

As shown in our discussion above, $p(\hat{\boldsymbol{x}}|\boldsymbol{x}) \simeq \mathcal{N}(\hat{\boldsymbol{x}}; \boldsymbol{x}, (\beta/2)\boldsymbol{I}_m)$ will hold in the metric defined space from our VAE analysis. This result is equivalent to Eq. 31 in their work if $D(\boldsymbol{x}, \hat{\boldsymbol{x}})$ is SSE and $\beta'$ is set to $\beta^{-1}$, as follows:

$$p(\hat{\boldsymbol{x}}|\boldsymbol{x}) \propto \exp(-\beta' D(\boldsymbol{x}, \hat{\boldsymbol{x}})) = \exp\left(-\frac{||\boldsymbol{x} - \hat{\boldsymbol{x}}||_2^2}{2(\beta/2)}\right) \propto \mathcal{N}(\hat{\boldsymbol{x}}; \boldsymbol{x}, (\beta/2)\boldsymbol{I}_m). \tag{32}$$

If $D(\boldsymbol{x}, \hat{\boldsymbol{x}})$ is not SSE, the use of the space transformation explained in appendix A.1 will lead to the same result.

## A.3   RELATION TO $\beta$-VAE (HIGGINS ET AL., 2017)

In the $\beta$-VAE work by Higgins et al. (2017), it is presumed that the objective $L_{\boldsymbol{x}}$ was not mistakenly distinguished from ELBO. In their work, ELBO equation is modified as:

$$E_{p(\boldsymbol{x})}[\,E_{\hat{\boldsymbol{x}} \sim p_\phi(\hat{\boldsymbol{x}}|\boldsymbol{x})}[q_\theta(\boldsymbol{x}|\hat{\boldsymbol{x}})] - \beta D_{\mathrm{KL}}(\cdot)\,]. \tag{33}$$

However, they use the predetermined probabilities of $p_\theta(\hat{\boldsymbol{x}}|\boldsymbol{x})$ such as the Bernoulli and Gaussian distributions in training (described in table 1 in Higgins et al. (2017)). As shown in our appendix H.2, the log-likelihoods of the Bernoulli and Gaussian distributions can be regarded as BCE and SSE metrics, respectively. As a result, the actual objective for training in Higgins et al. (2017) is not Eq. 33, but the objective $L_{\boldsymbol{x}}$ in Eq. 3 using BCE and SSE metrics with varying $\beta$. Thus ELBO as Eq. 1 form will become $\log p(\boldsymbol{x})$ in the BCE / SSE metric defined space regardless $\beta = 1$ or not, as shown in appendix A.1.

Actually, the equation 33 dose not show the log-likelihood of $\boldsymbol{x}$. When $D_{\mathrm{KL}}(\cdot) \simeq -\log p(\boldsymbol{x}) - (n/2)\log(\beta\pi e)$ and $E_{\hat{\boldsymbol{x}} \sim p(\hat{\boldsymbol{x}}|\boldsymbol{x})}[p(\boldsymbol{x}|\hat{\boldsymbol{x}})] \simeq -(n/2)\log(\beta\pi e)$ are applied, the value of Eq. 33 is derived as $\beta \log p(\boldsymbol{x}) + (\beta - 1)(n/2)\log(\beta\pi e)$, which is different from the log-likelihood of $\boldsymbol{x}$ if $\beta \neq 1$.

Correctly, what $\beta$-VAE really does is only to scale the variance of the pre-determined conditional distribution in the original VAE by a factor of $\beta$. In the case the pre-determined conditional distribution is Gaussian $\mathcal{N}(\boldsymbol{x}; \hat{\boldsymbol{x}}, \sigma^2\boldsymbol{I})$, the objective of $\beta$ can be can be rewritten as a linearly scaled original VAE objective with a Gaussian $\mathcal{N}(\boldsymbol{x}; \hat{\boldsymbol{x}}, \beta\sigma^2\boldsymbol{I})$:

$$
\begin{aligned}
E_{q_\phi(\cdot)}[\log \mathcal{N}(\boldsymbol{x}; \hat{\boldsymbol{x}}, \sigma^2\boldsymbol{I})] - \beta D_{\mathrm{KL}}(\cdot) &= E_{q_\phi(\cdot)}\left[-\frac{1}{2}\log 2\pi\sigma^2 - \frac{|\boldsymbol{x} - \hat{\boldsymbol{x}}|_2^2}{2\sigma^2}\right] - \beta D_{\mathrm{KL}}(\cdot) \\
&= \beta\left(E_{q_\phi(\cdot)}\left[-\frac{1}{2}\log 2\pi\beta\sigma^2 - \frac{|\boldsymbol{x} - \hat{\boldsymbol{x}}|_2^2}{2\beta\sigma^2}\right] - D_{\mathrm{KL}}(\cdot)\right) \\
&\quad + \frac{\beta}{2}\log 2\pi\beta\sigma^2 - \frac{1}{2}\log 2\pi\sigma^2 \\
&= \beta\left(E_{q_\phi(\cdot)}[\log \mathcal{N}(\boldsymbol{x}; \hat{\boldsymbol{x}}, \beta\sigma^2\boldsymbol{I})] - D_{\mathrm{KL}}(\cdot)\right) + \mathrm{const.}
\end{aligned} \tag{34}
$$

## A.4   RELATION TO ALEMI ET AL. (2018)

Alemi et al. (2018) discuss the rate-distortion trade-off by the theoretical entropy analysis. Their work is also presumed that the objective $L_{\boldsymbol{x}}$ was not mistakenly distinguished from ELBO, which leads to the incorrect discussion. In their work, the differential entropy for the input $H$, distortion $D$, and rate $R$ are derived carefully. They suggest that VAE with $\beta = 1$ is sensitive (unstable) because $D$ and $R$ can be arbitrary value on the line $R = H - \beta D = H - D$. Furthermore, they also suggest

that $R \geq H$, $D = 0$ at $\beta \to 0$ and $R = 0$, $D \geq H$ at $\beta \to \infty$ will hold as shown the figure 1 of their work.

In this appendix, we will show that $\beta$ determines the value of $R$ and $D$ specifically. We also show that $R \simeq H - D$ will hold regardless $\beta = 1$ or not.

In their work, these values of $H$, $D$, and $D$ are mathematically defined as:

$$H \equiv -\int \mathrm{d}\boldsymbol{x}\, p^*(\boldsymbol{x}) \log p^*(\boldsymbol{x}), \tag{35}$$

$$D \equiv -\int \mathrm{d}\boldsymbol{x}\, p^*(\boldsymbol{x}) \int \mathrm{d}\boldsymbol{z}\, e(\boldsymbol{z}|\boldsymbol{x}) \log d(\boldsymbol{x}|\boldsymbol{z}), \tag{36}$$

$$R \equiv \int \mathrm{d}\boldsymbol{x}\, p^*(\boldsymbol{x}) \int \mathrm{d}\boldsymbol{z}\, e(\boldsymbol{z}|\boldsymbol{x}) \log \frac{e(\boldsymbol{z}|\boldsymbol{x})}{m(\boldsymbol{z})}. \tag{37}$$

Here, $p^*(\boldsymbol{x})$ is a true PDF of $\boldsymbol{x}$, $e(\boldsymbol{z}|\boldsymbol{x})$ is a stochastic encoder, $e(\boldsymbol{z}|\boldsymbol{x})$ is a decoder, and $m(\boldsymbol{z})$ is a marginal probability of $\boldsymbol{z}$.

Our work allows a rough estimation of Eqs. 35-37 with $\beta$ by introducing the implicit isometric variable $\boldsymbol{y}$ as explained in our work.

Using isometric variable $\boldsymbol{y}$ and the relation $\mathrm{d}\boldsymbol{z}\, e(\boldsymbol{z}|\boldsymbol{x}) = \mathrm{d}\boldsymbol{y}\, e(\boldsymbol{y}|\boldsymbol{x})$, Eq. 36 can be rewritten as:

$$D = -\int \mathrm{d}\boldsymbol{x}\, p^*(\boldsymbol{x}) \int \mathrm{d}\boldsymbol{y}\, e(\boldsymbol{y}|\boldsymbol{x}) \log d(\boldsymbol{x}|\boldsymbol{y}). \tag{38}$$

Let $\boldsymbol{\mu}_y$ be the implicit isometric latent variable corresponding to the mean of encoder output $\boldsymbol{\mu}_{(\boldsymbol{x})}$. As discussed in section 3.3, $e(\boldsymbol{y}|\boldsymbol{x}) = \mathcal{N}(\boldsymbol{y}; \boldsymbol{\mu_y}, (\beta/2)\boldsymbol{I}_n)$ will hold. Because of isometricity, the value of $d(\boldsymbol{x}|\boldsymbol{y})$ will be also close to $e(\boldsymbol{y}|\boldsymbol{x}) = \mathcal{N}(\boldsymbol{y}; \boldsymbol{\mu_y}, (\beta/2)\boldsymbol{I}_n)$. Though $d(\boldsymbol{x}|\boldsymbol{z})$ must depend on $e(\boldsymbol{z}|\boldsymbol{x})$, this important point has not been discussed well in this work. By using the implicit isometric variable, we can connect both theoretically. Thus, $D$ can be estimated as:

$$\begin{aligned} D &\simeq \int \mathrm{d}\boldsymbol{x}\, p^*(\boldsymbol{x}) \int \mathrm{d}\boldsymbol{y}\, \mathcal{N}(\boldsymbol{y}; \boldsymbol{\mu}_y, (\beta/2)\boldsymbol{I}_n) \log \mathcal{N}(\boldsymbol{y}; \boldsymbol{\mu_y}, (\beta/2)\boldsymbol{I}_n) \\ &\simeq \int \mathrm{d}\boldsymbol{x}\, p^*(\boldsymbol{x}) \left( \frac{n}{2} \log(\beta\pi e) \right) \\ &= \frac{n}{2} \log(\beta\pi e) \end{aligned} \tag{39}$$

Second, $R$ is examined. $m(\boldsymbol{y})$ is a marginal probability of $\boldsymbol{y}$. Using the relation $\mathrm{d}\boldsymbol{z}\, e(\boldsymbol{z}|\boldsymbol{x}) = \mathrm{d}\boldsymbol{y}\, e(\boldsymbol{y}|\boldsymbol{x})$ and $e(\boldsymbol{z}|\boldsymbol{x})/m(\boldsymbol{z}) = (e(\boldsymbol{y}|\boldsymbol{x})(\mathrm{d}\boldsymbol{y}/\mathrm{d}\boldsymbol{z}))/(m(\boldsymbol{y})(\mathrm{d}\boldsymbol{y}/\mathrm{d}\boldsymbol{z})) = e(\boldsymbol{y}|\boldsymbol{x})/m(\boldsymbol{y})$, Eq. 37 can be rewritten as:

$$R \simeq \int \mathrm{d}\boldsymbol{x}\, p^*(\boldsymbol{x}) \int \mathrm{d}\boldsymbol{y}\, e(\boldsymbol{y}|\boldsymbol{x}) \log \frac{e(\boldsymbol{y}|\boldsymbol{x})}{m(\boldsymbol{y})}. \tag{40}$$

Because of isometricity, $e(\boldsymbol{y}|\boldsymbol{x}) \simeq p(\hat{\boldsymbol{x}}|\boldsymbol{x}) \simeq \mathcal{N}(\hat{\boldsymbol{x}}; \boldsymbol{x}, (\beta/2)\boldsymbol{I}_m)$ will approximately hold where $\hat{\boldsymbol{x}}$ denotes a decoder output. Thus $m(\boldsymbol{y})$ can be approximated by:

$$m(\boldsymbol{y}) \simeq \int \mathrm{d}\boldsymbol{x}\, p^*(\boldsymbol{x}) e(\boldsymbol{y}|\boldsymbol{x}) \simeq \int \mathrm{d}\boldsymbol{x}\, p^*(\boldsymbol{x})\, \mathcal{N}(\hat{\boldsymbol{x}}; \boldsymbol{x}, (\beta/2)\boldsymbol{I}_m) \tag{41}$$

Here, if $\beta/2$, i.e., added noise, is small enough compared to the variance of $\boldsymbol{x}$, a normal distribution function term in this equation will act like a delta function. Thus $m(\boldsymbol{y})$ can be approximated as:

$$m(\boldsymbol{y}) \simeq \int \mathrm{d}\acute{\boldsymbol{x}}\, p^*(\acute{\boldsymbol{x}})\, \delta(\acute{\boldsymbol{x}} - \boldsymbol{x}) \simeq p^*(\boldsymbol{x}). \tag{42}$$

In the similar way, the following approximation will also hold.

$$\int \mathrm{d}\boldsymbol{y}\, e(\boldsymbol{y}|\boldsymbol{x}) \log m(\boldsymbol{y}) \simeq \int \mathrm{d}\boldsymbol{y}\, e(\boldsymbol{y}|\boldsymbol{x}) \log p^*(\boldsymbol{x}) \simeq \int \mathrm{d}\acute{\boldsymbol{x}}\, \delta(\acute{\boldsymbol{x}} - \boldsymbol{x})\, \log\, p^*(\acute{\boldsymbol{x}}) \simeq \log p^*(\boldsymbol{x}) \tag{43}$$

By using these approximation and applying Eqs. 38-39, $R$ in Eq. 37 can be approximated as:

$$
\begin{aligned}
R &\simeq \int \mathrm{d}\boldsymbol{x}\, p^*(\boldsymbol{x}) \int \mathrm{d}\boldsymbol{y}\, e(\boldsymbol{y}|\boldsymbol{x}) \log \frac{e(\boldsymbol{y}|\boldsymbol{x})}{p^*(\boldsymbol{x})} \\
&\simeq -\int \mathrm{d}\boldsymbol{x}\, p^*(\boldsymbol{x})\, \log p^*(\boldsymbol{x}) - \left(-\int \mathrm{d}\boldsymbol{x}\, p^*(\boldsymbol{x}) \int \mathrm{d}\boldsymbol{y}\, e(\boldsymbol{y}|\boldsymbol{x}) \log e(\boldsymbol{y}|\boldsymbol{x})\right) \\
&\simeq H - \frac{n}{2}\log(\beta\pi e) \\
&\simeq H - D
\end{aligned}
\tag{44}
$$

As discussed above, $R$ and $D$ can be specifically derived from $\beta$. In addition, Shannon lower bound discussed in Alemi et al. (2018) can be roughly verified in the optimized VAE with clearer notations using $\beta$.

From the discussion above, we presume Alemi et al. (2018) might wrongly treat $D$ in their work. They suggest that VAE with $\beta = 1$ is sensitive (unstable) because $D$ and $R$ can be arbitrary value on the line $R = H - \beta D = H - D$; however, our work as well as Tishby et al. (1999) (appendix A.2) and Dai & Wipf (2019)(appendix A.5) show that the differential entropy of the distortion and rate, i.e., $D$ and $R$, are specifically determined by $\beta$ after optimization, and $R = H - D$ will hold for any $\beta$ regardless $\beta = 1$ or not. Alemi et al. (2018) also suggest $D$ should satisfy $D \geq 0$ because $D$ is a distortion; however, we suggest $D$ should be treated as a differential entropy and can be less than 0 because $\boldsymbol{x}$ is once handled as a continuous signal with a stochastic process in Eqs. 35-37. Here, $D \simeq (n/2)\log(\beta\pi e)$ can be $-\infty$ if $\beta \to 0$, as also shown in Dai & Wipf (2019). Thus, upper bound of $R$ at $\beta \to 0$ is not $H$, but $R = H - (-\infty) = \infty$, as shown in RD theory for a continuous signal. Huang et al. (2020) show this property experimentally in their figures 4-8 such that $R$ seems to diverge if MSE is close to 0.

## A.5 Relation to Dai et al. (2018) and Dai & Wipf (2019)

Our work is consistent with Dai et al. (2018) and Dai & Wipf (2019).

Dai et al. (2018) analyses VAE by assuming a linear model. As a result, the estimated posterior is constant. If the distribution of the manifold is the Gaussian, our work and Dai et al. (2018) give a similar result with constant posterior variances. For non-Gaussian data, however, the quantitative analysis such as probability estimation is intractable using their linear model. Our work reveals that the posterior variance gives a scaling factor between $\boldsymbol{z}$ in VAE and $\boldsymbol{y}$ in the isometric space when VAE is ideally trained with rich parameters. This is validated by Figures 3c and 3d, where the estimation of the posterior variance at each data point is a key.

Next, the relation to Dai & Wipf (2019) is discussed. They analyse a behavior of VAE when ideally trained. For example, the theorem 5 in their work shows that $D \to (d/2)\log\gamma + O(1)$ and $R \to -(\hat{\gamma}/2)\log\gamma + O(1)$ hold if $\gamma \to +0$, where $\gamma, d,$ and $\hat{\gamma}$ denote a variance of $d(\boldsymbol{x}|\boldsymbol{z})$, data dimension, and latent dimension, respectively. By setting $\gamma = \beta/2$ and $d = \hat{\gamma} = n$, this theorem is consistent with $R$ and $D$ derived in Eq. 39 and Eq. 44.

## A.6 Relation to transform coding (Goyal, 2001)

We show the optimum condition of VAE shown in Eq. 14 can be mapped to the optimum condition of transform coding (Goyal, 2001) as shown in Eq. 4. First, the derivation of Eq. 4 is explained by solving the optimal distortion assignment to each dimension. In the transform coding for $m$ dimensional the Gaussian data, an input data $\boldsymbol{x}$ is transformed to $\boldsymbol{z}$ using an orthonormal transform such as KLT/DCT. Then each dimensional component $z_j$ is encoded with allowing distortion $d_j$. Let $D$ be a target distortion satisfying $D = \sum_{j=1}^{m} d_j$. $\sigma_{zj}^2$ denotes a variance of each dimensional component $z_j$ for the input dataset. Then, a rate $R$ can be derived as $\sum_{j=1}^{m} \frac{1}{2}\log(\sigma_{zj}^2/d_j)$. By introducing a Lagrange parameter $\lambda$ and minimizing a rate-distortion optimization cost $L = D + \lambda R$, the optimum condition is derived as:

$$
\lambda_{\mathrm{opt}} = 2D/m, \quad d_j = D/m = \lambda_{\mathrm{opt}}/2. \tag{45}
$$

This result is consistent with Eq. 14 by setting $\beta = \lambda_{\mathrm{opt}} = 2D/m$. This implies that $L'_{\mathrm{G}}$ in Eq. 14 is a rate-distortion optimization (RDO) cost of transform coding when $\boldsymbol{x}$ is deterministically transformed to $\boldsymbol{y}$ in the implicit isometric space and stochastically encoded with a distortion $\beta/2$.

## B   ESTIMATION OF THE CODING LOSS AND TRANSFORM LOSS IN 1-DIMENSIONAL LINEAR VAE

This appendix estimates the coding loss and transform loss in 1-dimensional linear $\beta$-VAE for the Gaussian data, and also shows that the result is consistent with the Wiener filter. Let $x$ be a one dimensional data with the normal distribution:

$$x \in \mathbb{R}, \quad x \sim \mathcal{N}(x; 0, \sigma_x{}^2) \tag{46}$$

Let $z$ be a one dimensional latent variable. Following two linear encoder and decoder are provided with constant parameters $a$, $b$, and $\sigma_z$ to optimize:

$$\begin{aligned} z &= ax + \sigma_z \epsilon \ \text{ where } \ \epsilon \sim \mathcal{N}(\epsilon; 0, 1), \\ \hat{x} &= bz. \end{aligned} \tag{47}$$

First, KL divergence at x, $D_{\mathrm{KL}x}$ is derived. Due to the above relationship, we have

$$p(z) = \mathcal{N}(z; 0, (a\sigma_x)^2). \tag{48}$$

Using Eq. 6, KL-divergence at x can be evaluated as:

$$D_{\mathrm{KL}x} = -\log(\sigma_z p(z)) - \frac{1}{2}\log 2\pi e = -\log \sigma_x + \frac{a^2 x^2}{2} - \frac{1}{2}. \tag{49}$$

Second, the reconstruction loss at x $D_x$ is evaluated as:

$$D_x = \mathop{E}_{\epsilon \sim \mathcal{N}(\epsilon; 0, 1)}[(x - (b(ax + \sigma_z \epsilon)))^2] = ((ab - 1)x)^2 + b^2 \sigma_z{}^2. \tag{50}$$

Then, the loss objective $L_x = D_x + \beta D_{\mathrm{KL}x}$ is averaged over $x \sim \mathcal{N}(x; 0, \sigma_x{}^2)$, and the objective $L$ to minimize is derived as:

$$L = \mathop{E}_{x \sim \mathcal{N}(x; 0, \sigma_x{}^2)}[L_x] = (ab - 1)^2 \sigma_x{}^2 + b^2 \sigma_z{}^2 + \beta \left(-\log \sigma_z + \frac{a^2 \sigma_x{}^2}{2} - \frac{1}{2}\right). \tag{51}$$

Here, $(ab - 1)^2 \sigma_x{}^2$ and $b^2 \sigma_z{}^2$ in the last equation are corresponding to the transform loss $D_{\mathrm{T}}$ and coding loss $D_{\mathrm{C}}$, respectively.

By solving $\mathrm{d}L/\mathrm{d}a = 0$, $\mathrm{d}L/\mathrm{d}b = 0$, and $\mathrm{d}L/\mathrm{d}\sigma_z = 0$, $a$, $b$, and $\sigma_z$ are derived as follows:

$$\begin{aligned} a &= 1/\sigma_x, \\ b &= \frac{\sigma_x \left(1 + \sqrt{1 - 2\beta/\sigma_x{}^2}\right)}{2}, \\ \sigma_z &= \frac{2\sqrt{\beta/2}}{\sigma_x \left(1 + \sqrt{1 - 2\beta/\sigma_x{}^2}\right)}. \end{aligned} \tag{52}$$

From Eq. 52, $D_{\mathrm{T}}$ and $D_{\mathrm{C}}$ are derived as:

$$\begin{aligned} D_{\mathrm{T}} &= \left(\frac{\sqrt{1 - 2\beta/\sigma_x{}^2} - 1}{2}\right)^2 \sigma_x{}^2, \\ D_{\mathrm{C}} &= \beta/2. \end{aligned} \tag{53}$$

As shown in section 3.3, the added noise, $\beta/2$, should be reasonably smaller than the data variance $\sigma_x{}^2$. If $\sigma_x{}^2 \gg \beta$, $b$ and $\sigma_z$ in Eq. 52 can be approximated as:

$$D_{\mathrm{T}} \simeq \frac{(\beta/2)^2}{\sigma_x{}^2} = \frac{\beta/2}{\sigma_x{}^2} D_{\mathrm{C}}. \tag{54}$$

As shown in this equation, $D_{\mathrm{T}}/D_{\mathrm{C}}$ is small in the VAE where the added noise is reasonably small, and $D_{\mathrm{T}}$ can be ignored.

Next, the relation to the Wiener filter is discussed. We consider an simple 1-dimensional Gaussian process. Let $x \sim \mathcal{N}(x; 0, \sigma_x^2)$ be input data. Then, $x$ is scaled by $s$, and a Gaussian noise $n \sim$

$\mathcal{N}(n; 0, \sigma_n^2)$ is added. Thus, $y = s\,x + n$ is observed. From the Wiener filter theory, the estimated value with minimum distortion, $\hat{x}$ can be formulated as:

$$\hat{x} = \frac{s\sigma_x{}^2}{s^2\sigma_x{}^2 + \sigma_n{}^2} y. \tag{55}$$

In this case, the estimation error is derived as:

$$E[(\hat{x} - x)^2] = \frac{\sigma_n{}^4}{(s^2\sigma_x{}^2 + \sigma_n{}^2)^2}\sigma_x{}^2 + \frac{s^2\sigma_x{}^4}{(s^2\sigma_x{}^2 + \sigma_n{}^2)^2}\sigma_n{}^2 = \frac{\sigma_x{}^2}{\sigma_x{}^2 + (\sigma_n{}^2/s^2)}(\sigma_n{}^2/s^2). \tag{56}$$

In the second equation, the first term is corresponding to the transform loss, and the second term is corresponding to the coding loss. Here the ratio of the transform loss and coding loss is derived as $\sigma_n{}^2/(s^2\sigma_x{}^2)$. By appying $s = 1/\sigma_x$ and $\sigma_n = \sigma_z$ to $\sigma_n{}^2/(s^2\sigma_x{}^2)$ and assuming $\sigma_x^2 \gg \beta/2$, this ratio can be described as:

$$\frac{\sigma_n{}^2}{s^2\sigma_x{}^2} = \sigma_z{}^2 = \frac{\beta/2}{\sigma_x^2} \frac{4}{\left(1 + \sqrt{1 - 2\beta/\sigma_x{}^2}\right)^2} = \frac{\beta/2}{\sigma_x^2} + O\left(\left(\frac{\beta/2}{\sigma_x^2}\right)^2\right). \tag{57}$$

This result is consistent with Eq. 54, implying that optimized VAE and the Wiener filter show similar behaviours.

## C  DERIVATION OF QUANTITATIVE PROPERTIES IN SECTION 3.5

### C.1  DERIVATION OF THE ESTIMATED VARIANCE

This appendix explains the derivation of Eq. 21 in Section 3.5. Here, we assume that $z_j$ is mapped to $y_j$ such that $y_j$ is set to 0 at $z_j = 0$. We also assume that the prior distribution is $\mathcal{N}(\boldsymbol{z}; 0, \boldsymbol{I}_n)$. The variance is derived by the subtraction of $E[y_j]^2$, the square of the mean, from $E[y_j^2]$, the square mean. Thus, the approximations of both $E[y_j]$ and $E[y_j^2]$ are needed.

First, the approximation of the mean $E[y_j]$ is explained. Because the cumulative distribution functions (CDFs) of $y_j$ are the same as CDF of $z_j$, the following equations hold:

$$\int_{-\infty}^0 p(y_j)\mathrm{d}y_j = \int_{-\infty}^0 p(z_j)\mathrm{d}z_j = 0.5, \quad \int_0^\infty p(y_j)\mathrm{d}y_j = \int_0^\infty p(z_j)\mathrm{d}z_j = 0.5. \tag{58}$$

This equation means that the median of the $y_j$ distribution is 0. Because the mean and median are close in most cases, the mean $E[y_j]$ can be approximated as 0. As a result, the variance of $y_j$ can be approximated by the square mean $E[y_j^2]$.

Second, the approximation of the square mean $E[y_j^2]$ is explained. The standard deviation of the posterior $\sigma_{j(\boldsymbol{x})}$ is assumed as a function of $z_j$, regardless of $\boldsymbol{x}$. This function is denoted as $\sigma_j(z_j)$. For $z_j \geq 0$, $y_j$ is approximated as follows, using Eq. 11 and replacing the average of $1/\sigma_j(\acute{z}_j)$ over $\acute{z}_j = [0, z_j]$ by $1/\sigma_j(z_j)$:

$$y_j = \int_0^{z_j} \frac{\mathrm{d}y_j}{\mathrm{d}\acute{z}_j}\mathrm{d}\acute{z}_j = \sqrt{\frac{\beta}{2}}\int_0^{z_i}\frac{1}{\sigma_j(\acute{z}_j)}\mathrm{d}\acute{z}_i \simeq \sqrt{\frac{\beta}{2}}\frac{1}{\sigma_j(z_j)}\int_0^{z_j}\mathrm{d}\acute{z}_j = \sqrt{\frac{\beta}{2}}\frac{z_j}{\sigma_j(z_j)}. \tag{59}$$

The same approximation is applied to $z_i < 0$. Then the square mean of $y_i$ is approximated as follows, assuming that the correlation between $\sigma(z_j)^{-2}$ and $z_j{}^2$ is low:

$$\int y_j{}^2 p(y_j)\mathrm{d}y_j \simeq \frac{\beta}{2}\int \left(\frac{z_j}{\sigma_j(z_j)}\right)^2 p(z_j)\mathrm{d}z_j \simeq \frac{\beta}{2}\int \sigma_j(z_j)^{-2}p(z_j)\mathrm{d}z_j \int z_j{}^2 p(z_j)\mathrm{d}z_j. \tag{60}$$

Finally, the square mean of $y_i$ is approximated as the following equation, using $\int z_j{}^2 p(z_j)\mathrm{d}z_j = 1$ and replacing $\sigma_j(z_j)^2$ by $\sigma_{j(\boldsymbol{x})}{}^2$, i.e., the posterior variance derived from the input data:

$$\int y_j{}^2 p(y_j)\mathrm{d}y_j \simeq \frac{\beta}{2}\int \sigma_j(z_j)^{-2}p(z_j)\mathrm{d}z_j \simeq \frac{\beta}{2}\mathop{E}_{z_j \sim p(z_j)}[\sigma_j(z_j)^{-2}] \simeq \frac{\beta}{2}\mathop{E}_{\boldsymbol{x} \sim p(\boldsymbol{x})}[\sigma_{j(\boldsymbol{x})}{}^{-2}]. \tag{61}$$

Although some rough approximations are used in the expansion, the estimated variance in the last equation seems still reasonable, because $\sigma_{j(\boldsymbol{x})}$ shows a scale factor between $y_j$ and $z_j$ while the variance of $z_j$ is always 1 for the prior $\mathcal{N}(z_j; 0, 1)$. Considering the variance of the prior $\int z_j{}^2 p(z_j)\mathrm{d}z_j$ in the expansion, this estimation method can be applied to any prior distribution.

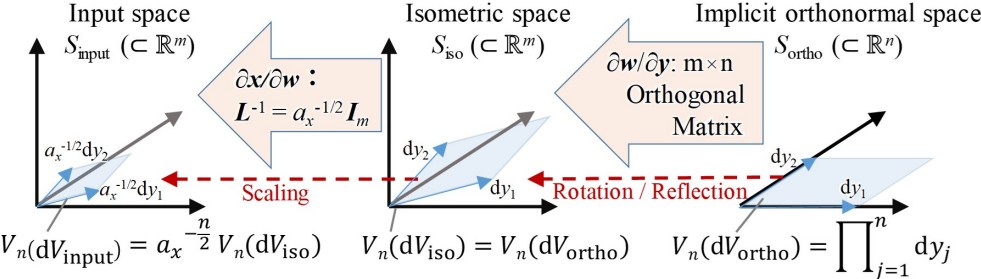

Figure 7: Projection of the volume element from the implicit orthonormal space to the isometric space and input space. $V_n(\cdot)$ denotes $n$-dimensional volume.

### C.2 DERIVATION OF THE DATA PROBABILITY ESTIMATION

This appendix shows the derivation of variables in Eqs. 22 and 23. First, the derivation of $L_{\boldsymbol{x}}$ for the input $\boldsymbol{x}$ is described. Then, the PDF ratio between the input space and inner product space is explained for the cases $m = n$ and $m > n$.

**Derivation of $L_{\boldsymbol{x}}$ for the input $\boldsymbol{x}$ :**
As shown in in Eq. 1, $L_{\boldsymbol{x}}$ is denoted as $-E_{\boldsymbol{z} \sim q_\phi(\boldsymbol{z}|\boldsymbol{x})}[\,\cdot\,] + \beta D_{\mathrm{KL}}(\,\cdot\,)$. We approximate $E_{\boldsymbol{z} \sim q_\phi(\boldsymbol{z}|\boldsymbol{x})}[\,\cdot\,]$ as $\frac{1}{2}(D(\boldsymbol{x}, \mathrm{Dec}_\theta(\boldsymbol{\mu_x} + \boldsymbol{\sigma_x})) + D(\boldsymbol{x}, \mathrm{Dec}_\theta(\boldsymbol{\mu_x} - \boldsymbol{\sigma_x})))$, i.e., the average of two samples, instead of the average over $\boldsymbol{z} \sim q_\phi(\boldsymbol{z}|\boldsymbol{x})$. $D_{\mathrm{KL}}(\,\cdot\,)$ can be calculated from $\boldsymbol{\mu_x}$ and $\boldsymbol{\sigma_x}$ using Eq. 2.

**The PDF ratio in the case $m = n$:**
The PDF ratio for $m = n$ is a Jacobian determinant between two spaces. First, $\left(\frac{\partial \boldsymbol{x}}{\partial \boldsymbol{y}}\right)^T \boldsymbol{G_x}\left(\frac{\partial \boldsymbol{x}}{\partial \boldsymbol{y}}\right) = \boldsymbol{I}_m$ holds from Eq. 12. $|\partial \boldsymbol{x}/\partial \boldsymbol{y}|^2 \, |\boldsymbol{G_x}| = 1$ also holds by calculating the determinant. Finally, $|\partial \boldsymbol{x}/\partial \boldsymbol{y}|$ is derived as $|\boldsymbol{G}_x|^{1/2}$ using $|\partial \boldsymbol{y}/\partial \boldsymbol{x}| = |\partial \boldsymbol{x}/\partial \boldsymbol{y}|^{-1}$.

**The PDF ratio in the case $m > n$ and $\boldsymbol{G_x} = a_{\boldsymbol{x}}\boldsymbol{I}_m$:**
Although the strict derivation needs the treatment of the Riemannian manifold, we provide a simple explanation in this appendix. Here, it is assumed that $D_{\mathrm{KL}(j)}(\cdot) > 0$ holds for all $j = [1, .. n]$. If $D_{\mathrm{KL}(j)}(\cdot) = 0$ for some $j$, $n$ is replaced by the number of latent variables with $D_{\mathrm{KL}(j)}(\cdot) > 0$.

For the implicit isometric space $S_{\mathrm{iso}}(\subset \mathbb{R}^m)$, there exists a matrix $\boldsymbol{L_x}$ such that both $\boldsymbol{y} = \boldsymbol{L_x}\boldsymbol{x}$ and $\boldsymbol{G_x} = {}^t\boldsymbol{L_x}\boldsymbol{L_x}$ holds. $\boldsymbol{w}$ denotes a point in $S_{\mathrm{iso}}$, i.e., $\boldsymbol{w} \in S_{\mathrm{iso}}$. Because $\boldsymbol{G_x}$ is assumed as $a_{\boldsymbol{x}}\boldsymbol{I}_m$ in Section 3.5, $\boldsymbol{L_x} = a_{\boldsymbol{x}}^{1/2}\boldsymbol{I}_m$ holds. Then, the mapping function $\boldsymbol{w} = h(\boldsymbol{x})$ between $S_{\mathrm{input}}$ and $S_{\mathrm{iso}}$ is defined, such that:

$$\frac{\partial h(\boldsymbol{x})}{\partial \boldsymbol{x}} = \frac{\partial \boldsymbol{w}}{\partial \boldsymbol{x}} = \boldsymbol{L_x}, \ \ \text{and} \ \ h(\boldsymbol{x}^{(0)}) = \boldsymbol{w}^{(0)} \ \ \text{for} \ \exists\, \boldsymbol{x}^{(0)} \in S_{\mathrm{input}} \ \ \text{and} \ \exists\, \boldsymbol{w}^{(0)} \in S_{\mathrm{iso}}. \tag{62}$$

Let $\delta\boldsymbol{x}$ and $\delta\boldsymbol{w}$ are infinitesimal displacements around $\boldsymbol{x}$ and $\boldsymbol{w} = h(\boldsymbol{x})$, such that $\boldsymbol{w} + \delta\boldsymbol{w} = h(\boldsymbol{x} + \delta\boldsymbol{x})$. Then the next equation holds from Eq. 62:

$$\delta\boldsymbol{w} = \boldsymbol{L_x}\delta\boldsymbol{x} \tag{63}$$

Let $\delta\boldsymbol{x}^{(1)}, \delta\boldsymbol{x}^{(2)}, \delta\boldsymbol{w}^{(1)}$, and $\delta\boldsymbol{w}^{(2)}$ be two arbitrary infinitesimal displacements around $\boldsymbol{x}$ and $\boldsymbol{w} = h(\boldsymbol{x})$, such that $\delta\boldsymbol{w}^{(1)} = \boldsymbol{L_x}\delta\boldsymbol{x}^{(1)}$ and $\delta\boldsymbol{w}^{(2)} = \boldsymbol{L_x}\delta\boldsymbol{x}^{(2)}$. Then the following equation holds, where $\cdot$ denotes the dot product.

$$ {}^t\delta\boldsymbol{x}^{(1)}\boldsymbol{G_x}\delta\boldsymbol{x}^{(2)} = {}^t(\boldsymbol{L_x}\delta\boldsymbol{x}^{(1)})(\boldsymbol{L_x}\delta\boldsymbol{x}^{(2)}) = \delta\boldsymbol{w}^{(1)} \cdot \delta\boldsymbol{w}^{(2)} \tag{64}$$

This equation shows the isometric mapping from the inner product space for $\boldsymbol{x} \in S_{\mathrm{input}}$ with the metric tensor $\boldsymbol{G_x}$ to the Euclidean space for $\boldsymbol{w} \in S_{\mathrm{iso}}$.

Note that all of the column vectors in the Jacobian matrix $\partial\boldsymbol{x}/\partial\boldsymbol{y}$ also have a unit norm and are orthogonal to each other in the metric space for $\boldsymbol{x} \in S_{\mathrm{input}}$ with the metric tensor $\boldsymbol{G_x}$. Therefore, the $m \times n$ Jacobian matrix $\partial\boldsymbol{w}/\partial\boldsymbol{y}$ should have a property that all of the column vectors have a unit norm and are orthogonal to each other in the Euclidean space.

Then $n$-dimensional space which is composed of the meaningful dimensions from the implicit isometric space is named as the implicit orthonormal space $S_{\text{ortho}}$. Figure 7 shows the projection of the volume element from the implicit orthonormal space to the isometric space and input space. Let $dV_{\text{ortho}}$ be an infinitesimal $n$-dimensional volume element in $S_{\text{ortho}}$. This volume element is a $n$-dimensional rectangular solid having each edge length $dy_j$. Let $V_n(dV_X)$ be the $n$-dimensional volume of a volume element $dV_X$. Then, $V_n(dV_{\text{ortho}}) = \prod_j^n dy_j$ holds. Next, $dV_{\text{ortho}}$ is projected to $n$ dimensional infinitesimal element $dV_{\text{iso}}$ in $S_{\text{iso}}$ by $\partial \boldsymbol{w}/\partial \boldsymbol{y}$. Because of the orthonormality, $dV_{\text{iso}}$ is equivalent to the rotation / reflection of $dV_{\text{ortho}}$, and $V_n(dV_{\text{iso}})$ is the same as $V_n(dV_{\text{ortho}})$, i.e., $\prod_j^n dy_j$. Then, $dV_{\text{iso}}$ is projected to $n$-dimensional element $dV_{\text{input}}$ in $S_{\text{input}}$ by $\partial \boldsymbol{x}/\partial \boldsymbol{w} = \boldsymbol{L}_{\boldsymbol{x}}^{-1} = a_{\boldsymbol{x}}^{-1/2}\boldsymbol{I}_m$. Because each dimension is scaled equally by the scale factor $a_{\boldsymbol{x}}^{-1/2}$, $V_n(dV_{\text{input}}) = \prod_j^n a_{\boldsymbol{x}}^{-1/2}dy_j = a_{\boldsymbol{x}}^{-n/2} V_n(dV_{\text{ortho}})$ holds. Here, the ratio of the volume element between $S_{\text{input}}$ and $S_{\text{ortho}}$ is $V_n(dV_{\text{input}})/V_n(dV_{\text{ortho}}) = a_{\boldsymbol{x}}^{-n/2}$. Note that the PDF ratio is derived by the reciprocal of $V_n(dV_{\text{input}})/V_n(dV_{\text{ortho}})$. As a result, the PDF ratio is derived as $a_{\boldsymbol{x}}^{n/2}$.

# D DETAILS OF THE NETWORKS AND TRAINING CONDITIONS IN THE EXPERIMENTS

This appendix explains the networks and training conditions in Section 4.

## D.1 TOY DATA SET

This appendix explains the details of the networks and training conditions in the experiment of the toy data set in Section 4.1.

**Network configurations:**
FC(i, o, f) denotes a FC layer with input dimension i, output dimension o, and activate function f.

The encoder network is composed of FC(16, 128, tanh)-FC(128, 64, tahh)-FC(64, 3, linear)×2 (for $\mu$ and $\sigma$). The decoder network is composed of FC(3, 64, tanh)-FC(64, 128, tahh)-FC(128, 16, linear).

**Training conditions:**
The reconstruction loss $D(\cdot, \cdot)$ is derived such that the loss per input dimension is calculated and all of the losses are averaged by the input dimension $m = 16$. The KL divergence is derived as a summation of $D_{\text{KL}(j)}(\cdot)$ as explained in Eq. 2.

In our code, we use essentially the same, but a constant factor scaled loss objective from the original $\beta$-VAE form $L_{\boldsymbol{x}} = D(\cdot, \cdot) + \beta D_{\text{KL}(j)}(\cdot)$ in Eq. 1, such as:

$$L_{\boldsymbol{x}} = \lambda D(\cdot, \cdot) + D_{\text{KL}(j)}(\cdot). \tag{65}$$

Equation 65 is essentially equivalent to $L = D(\cdot, \cdot) + \beta D_{\text{KL}(j)}(\cdot)$, multiplying a constant $\lambda = \beta^{-1}$ to the original form. The reason why we use this form is as follows. Let $\text{ELBO}_{\text{true}}$ be the true ELBO in the sense of log-likelihood, such as $E[\log p(\boldsymbol{x})]$. As shown in Section 3.3, the minimum of the loss objective in the original $\beta$-VAE form is likely to be a $-\beta\text{ELBO}_{\text{true}} + \text{Constant}$. If we use Eq. 65, the minimum of the loss objective will be $-\text{ELBO}_{\text{true}} + \text{Constant}$, which seems more natural form of ELBO. Thus, Eq. 65 allows estimating a data probability from $L_{\boldsymbol{x}}$ in Eqs. 22 and 23, without scaling $L_{\boldsymbol{x}}$ by $1/\beta$.

Then the network is trained with $\lambda = \beta^{-1} = 100$ using 500 epochs with a batch size of 128. Here, Adam optimizer is used with the learning rate of 1e-3. We use a PC with CPU Inter(R) Xeon(R) CPU E3-1280v5@3.70GHz, 32GB memory equipped with NVIDIA GeForce GTX 1080. The simulation time for each trial is about 20 minutes, including the statistics evaluation codes.

In our experiments, $\lambda$ or $\beta^{-1}$, i.e., 100, seems somewhat large. This is caused by the use of the mean square error as a reconstruction loss. In contrast, KL divergence is the sum for the whole image, which can be thought of as a rate for the whole image. Considering the number of input dimensions, $\beta' = (\lambda/16)^{-1} = 16/\lambda = 0.16$ is thought of as $\beta$ in the general form of VAE.

### D.2 CelebA data set

This appendix explains the details of the networks and training conditions in the experiment of the toy data set in Section 4.2.

**Network configurations:**
CNN(w, h, s, c, f) denotes a CNN layer with kernel size (w, h), stride size s, dimension c, and activate function f. GDN and IGDN [2] are activation functions designed for image compression (Ballé et al., 2016). This activation function is effective and popular in deep image compression studies.

The encoder network is composed of CNN(9, 9, 2, 64, GDN) - CNN(5, 5, 2, 64, GDN) - CNN(5, 5, 2, 64, GDN) - CNN(5, 5, 2, 64, GDN) - FC(1024, 1024, softplus) - FC(1024, 32, None)$\times$2 (for $\boldsymbol{\mu}$ and $\boldsymbol{\sigma}$) in encoder.

The decoder network is composed of FC(32, 1024, softplus) - FC(1024, 1024, softplus) - CNN(5, 5, 2, 64, IGDN) - CNN(5, 5, 2, 64, IGDN) - CNN(5, 5, 2, 64, IGDN)-CNN(9, 9, 2, 3, IGDN).

**Training conditions:**
In this experiment, SSIM explained in Appendix H.2 is used as a reconstruction loss. The reconstruction loss $D(\cdot, \cdot)$ is derived as follows. Let SSIM be a SSIM calculated from two input images. Then $1 - \text{SSIM}$ is set to $D(\cdot, \cdot)$. The KL divergence is derived as a summation of $D_{\text{KL}(j)}(\cdot)$ as explained in Eq. 2.

We also use the loss form as in Equation 65 in our code. In the case of the decomposed loss, the loss function $L_{\boldsymbol{x}}$ is set to $\lambda(D(\boldsymbol{x}, \breve{\boldsymbol{x}}) + D(\breve{\boldsymbol{x}}, \hat{\boldsymbol{x}})) + D_{\text{KL}}(\cdot)$ in our code. Then, the network is trained with $\lambda = \beta^{-1} = 1,000$ using a batch size of 64 for 300,000 iterations. Here, Adam optimizer is used with the learning rate of 1e-3.

We use a PC with CPU Intel(R) Core(TM) i7-6850K CPU @ 3.60GHz, 12GB memory equipped with NVIDIA GeForce GTX 1080. The simulation time for each trial is about 180 minutes, including the statistics evaluation codes.

In our experiments, $\lambda = \beta^{-1} = 1,000$ seems large. This is caused by the use of SSIM. As explained in Appendix H.2, SSIM is measured for a whole image, and its range is between 0 and 1. The order of $1 - \text{SSIM}$ is almost equivalent to the mean square error per pixel, as shown in Eq. 74. As explained in Appendix D.1, KL divergence is thought of as a rate for the whole image. Considering the number of pixels in a image, $\beta' = (\lambda/(64 \times 64))^{-1} = 4096/\lambda = 4.096$ is comparable to $\beta$ in the general form of VAE.

---

[2]Google provides a code in the official Tensorflow library (https://github.com/tensorflow/compression)

# E  ADDITIONAL RESULTS IN THE TOY DATASETS

## E.1  SCATTERING PLOTS FOR THE SQUARE ERROR LOSS IN SECTION

4.1 Figure 8a shows the plots of $p(\boldsymbol{x})$ and estimated probabilities for the square error coding loss in Section 4.1, where the scale factor $a_{\boldsymbol{x}}$ in Eq. 23 is 1. Thus, both $\exp(-L_x/\beta)$ and $p(\boldsymbol{\mu}_{(\boldsymbol{x})})\prod_j \sigma_{j(\boldsymbol{x})}$ show a high correlation, allowing easy estimation of the data probability in the input space. In contrast, $p(\boldsymbol{\mu}_{(\boldsymbol{x})})$ still shows a low correlation. These results are consistent with our theory.

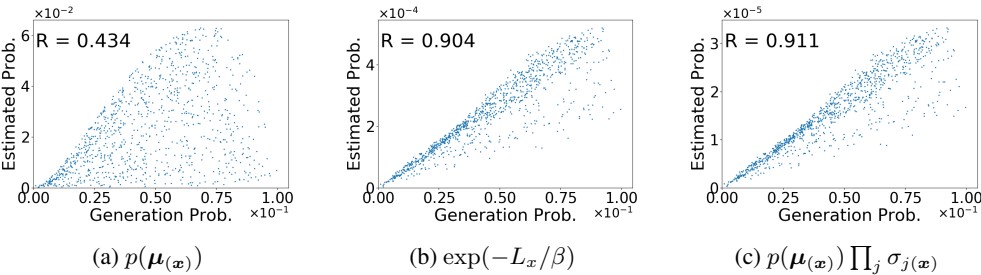

(a) $p(\boldsymbol{\mu}_{(\boldsymbol{x})})$         (b) $\exp(-L_x/\beta)$         (c) $p(\boldsymbol{\mu}_{(\boldsymbol{x})})\prod_j \sigma_{j(\boldsymbol{x})}$

Figure 8: Plots of the data generation probability (x-axis) versus estimated probabilities (y-axes) for the square error loss. y-axes are (a) $p(\boldsymbol{\mu}_{(\boldsymbol{x})})$, (b) $\exp(-L_x/\beta)$, and (c) $p(\boldsymbol{\mu}_{(\boldsymbol{x})})\prod_j \sigma_{j(\boldsymbol{x})}$.

## E.2  ABLATION STUDY USING 3 TOY DATASETS, 3 CODING LOSSES, AND 10 $\beta$ PARAMETERS.

In this appendix, we explain the ablation study for the toy datasets. We introduce three toy datasets and three coding losses including those used in Section 4.1. We also change $\beta^{-1} = \lambda$ from 1 to $1{,}000$ in training. The details of the experimental conditions are shown as follows.

**Datasets:** First, we call the toy dataset used in Section 4.1 the Mix dataset in order to distinguish three datasets. The second dataset is generated such that three dimensional variables $s_1$, $s_2$, and $s_3$ are sampled in accordance with the distributions $p(s_1)$, $p(s_2)$, and $p(s_3)$ in Figure 9. The variances of the variables are the same as those of the Mix dataset, i.e., 1/6, 2/3, and 8/3, respectively. We call this the Ramp dataset. Because the PDF shape of this dataset is quite different from the prior $\mathcal{N}(\boldsymbol{z}; 0, I_3)$, the fitting will be the most difficult among the three. The third dataset is generated such that three dimensional variables $s_1$, $s_2$, and $s_3$ are sampled in accordance with the normal distributions $\mathcal{N}(s_1; 0, 1/6)$, $\mathcal{N}(s_2; 0, 2/3)$, and $\mathcal{N}(s_3; 0, 8/3)$, respectively. We call this the Norm dataset. The fitting will be the easiest, because both the prior and input have the normal distributions, and the posterior standard deviation, given by the PDF ratio at the same CDF, can be a constant.

**Coding losses:** Two of the three coding losses is the square error loss and the downward-convex loss described in Section 4.1. The third coding loss is a upward-convex loss which we design as Eq. 66 such that the scale factor $a_{\boldsymbol{x}}$ becomes the reciprocal of the scale factor in Eq. 24:

$$D(\boldsymbol{x}, \hat{\boldsymbol{x}}) = a_{\boldsymbol{x}}\|\boldsymbol{x} - \hat{\boldsymbol{x}}\|_2^2, \quad \text{where } a_{\boldsymbol{x}} = (2/3 + 2\|\boldsymbol{x}\|_2^2/21)^{-1} \text{ and } \boldsymbol{G}_x = a_{\boldsymbol{x}}\boldsymbol{I}_m. \tag{66}$$

Figure 10 shows the scale factors $a_{\boldsymbol{x}}$ in Eqs. 24 and 66, where $s_1$ in $\boldsymbol{x} = (s_1, 0, 0)$ moves within $\pm 5$.

**Parameters:** As explained in Appendix D.1, $\lambda = 1/\beta$ is used as a hyper parameter. Specifically, $\lambda = 1, 2, 5, 10, 20, 50, 100, 200, 500$, and $1{,}000$ are used.

Figures 11 - 19 show the property measurements for all combinations of the datasets and coding losses, with changing $\lambda$. In each Figure, the estimated norms of the implicit transform are shown in the figure (a), the ratios of the estimated variances are shown in the figure (b), and the correlation coefficients between $p(\boldsymbol{x})$ and estimated data probabilities are shown in the figure (c), respectively.

First, the estimated norm of the implicit transform in the figures (a) is discussed. In all conditions, the norms are close to 1 as described in Eq. 20 in the $\lambda$ range 50 to 1000. These results show consistency with our theoretical analysis, supporting the existence of the implicit orthonormal transform. The values in the Norm dataset are the closest to 1, and those in the Ramp dataset are the most different, which seems consistent with the difficulty of the fitting.

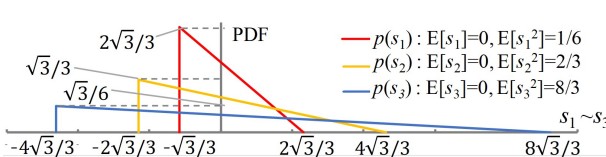
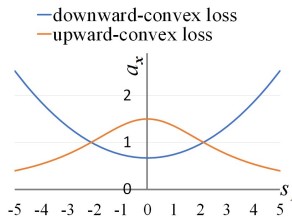

Figure 9: PDFs of three variables to generate a Ramp dataset.

Figure 10: Scale factor $a_{\boldsymbol{x}}$ for the downward-convex loss and upward-convex loss.

Second, the ratio of the estimated variances is discussed. In the figures (b), $\mathrm{Var}(z_j)$ denotes the estimated variance, given by the average of $\sigma_{j(\boldsymbol{x})}^{-2}$. Then, $\mathrm{Var}(z_2)/\mathrm{Var}(z_1)$ and $\mathrm{Var}(z_3)/\mathrm{Var}(z_1)$ are plotted. In all conditions, the ratios of $\mathrm{Var}(z_2)/\mathrm{Var}(z_1)$ and $\mathrm{Var}(z_3)/\mathrm{Var}(z_1)$ are close to the variance ratios of the input variables, i.e., 4 and 16, in the $\lambda$ range 5 to 500. Figure 20 shows the detailed comparison of the ratio for the three datasets and three coding losses at $\lambda = 100$. In most cases, the estimated variances in the downward-convex loss are the smallest, and those in the upward-convex loss are the largest, which is more distinct for $\mathrm{Var}(z_3)/\mathrm{Var}(z_1)$. This can be explained as follows. When using the downward-convex loss, the space region with a large norm is thought of as shrinking in the inner product space, as described in Section 4.1. This will make the variance smaller. In contrast, when using the upward-convex loss, the space region with a large norm is thought of as expanding in the inner product space, making the variance larger. Here, the dependency of the losses on the ratio changes is less in the Norm dataset. The possible reason is that data in the normal distribution concentrate around the center, having less effect on the loss scale factor in the downward-convex loss and upward-convex loss.

Third, the correlation coefficients between $p(\boldsymbol{x})$ and the estimated data probabilities in the figures (c) are discussed. In the Mix dataset and Ramp dataset, the correlation coefficients are around 0.9 in the $\lambda$ range from 20 to 200 when the estimated probabilities $a_{\boldsymbol{x}}{}^{n/2}p(\boldsymbol{\mu}_{(\boldsymbol{x})})\prod_{j=1}^{n}\sigma_{j(\boldsymbol{x})}$ and $a_{\boldsymbol{x}}{}^{n/2}\exp(-(1/\beta)L_{\boldsymbol{x}})$ in Eq. 23 are used. When using $p(\boldsymbol{\mu}_{(\boldsymbol{x})})\prod_{j=1}^{n}\sigma_{j(\boldsymbol{x})}$ and $\exp(-(1/\beta)L_{\boldsymbol{x}})$ in the downward-convex loss and upward-convex loss, the correlation coefficients become worse. In addition, when using the prior probability $p(\boldsymbol{\mu}_{(\boldsymbol{x})})$, the correlation coefficients always show the worst. In the Norm dataset, the correlation coefficients are close to 1.0 in the wider range of $\lambda$ when using the estimated distribution in Eq. 23. When using $p(\boldsymbol{\mu}_{(\boldsymbol{x})})\prod_{j=1}^{n}\sigma_{j(\boldsymbol{x})}$ and $\exp(-(1/\beta)L_{\boldsymbol{x}})$ in the downward-convex loss and upward-convex loss, the correlation coefficients also become worse. When using the prior probability $p(\boldsymbol{\mu}_{(\boldsymbol{x})})$, however, the correlation coefficients are close to 1 in contrast to the other two datasets. This can be explained because both the input distribution and the prior distribution are the same normal distribution, allowing the posterior variances almost constant. These results also show consistency with our theoretical analysis.

Figure 21 shows the dependency of the coding loss on $\beta$ for the Mix, Ramp, and Norm dataset using square the error loss. From $D_{\mathrm{G}}$ in Eq. 14 and $n = 3$, the theoretical value of coding loss is $\frac{3\beta}{2}$, as also shown in the figure. Unlike Figs. 11-19, $x$-axis is $\beta = \lambda^{-1}$ to evaluate the linearity. As expected in section 3.3, the coding losses are close to the theoretical value where $\beta < 0.1$, i.e., $\lambda > 10$.

Figure 22 shows the dependency of the ratio of transform loss to coding loss on $\beta$ for the Mix, Ramp, and Norm dataset using square the error loss. From Eq. 54, the estimated transform loss is $\sum_{i=1}^{3}(\beta/2)^2/\mathrm{Var}(s_i) = \frac{63\beta^2}{32}$. Thus the theoretical value is $(\frac{63\beta^2}{32})/(\frac{3\beta}{2}) = \frac{21\beta}{16}$, as is also shown in the figure. $x$-axis is also $\beta = \lambda^{-1}$ like Figure 21. Considering the correlation coefficient discussed above, the useful range of $\beta$ seems between 0.005-0.05 (20-200 for $\lambda$). In this range, the ratio is less than 0.1, implying the transform loss is almost negligible. As expected in section 3.2 and appendix B, the ratio is close to the theoretical value where $\beta > 0.01$, i.e., $\lambda < 100$. For $\beta < 0.01$, the conversion loss is still negligibly small, but the ratio is somewhat off the theoretical value. The reason is presumably that the transform loss is too small to fit the network.

As shown above, this ablation study strongly supports our theoretical analysis in sections 3.

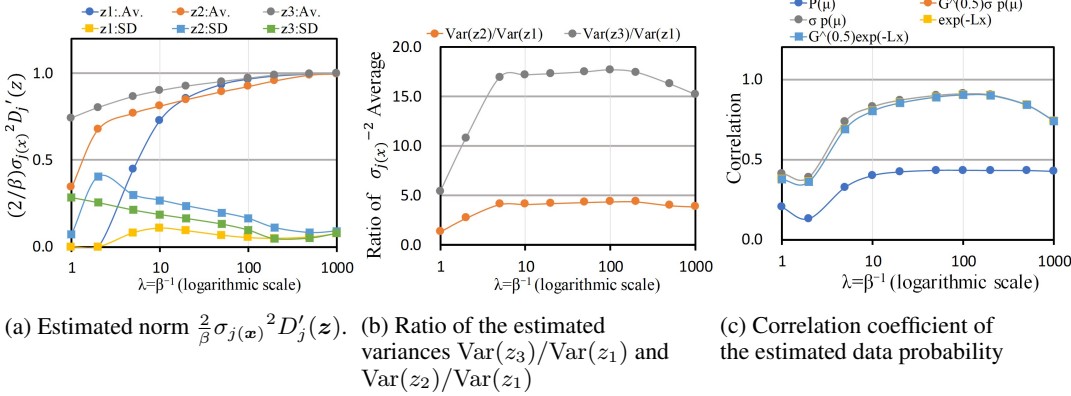

(a) Estimated norm $\frac{2}{\beta}\sigma_{j(\boldsymbol{x})}{}^2 D_j'(\boldsymbol{z})$.

(b) Ratio of the estimated variances $\mathrm{Var}(z_3)/\mathrm{Var}(z_1)$ and $\mathrm{Var}(z_2)/\mathrm{Var}(z_1)$

(c) Correlation coefficient of the estimated data probability

Figure 11: Property measurements of the Mix dataset using the square error loss. $\lambda$ is changed from 1 to 1,000. $\mathrm{Var}(z_j)$ denotes the estimated variance, given by the average of $\sigma_{j(\boldsymbol{x})}^{-2}$.

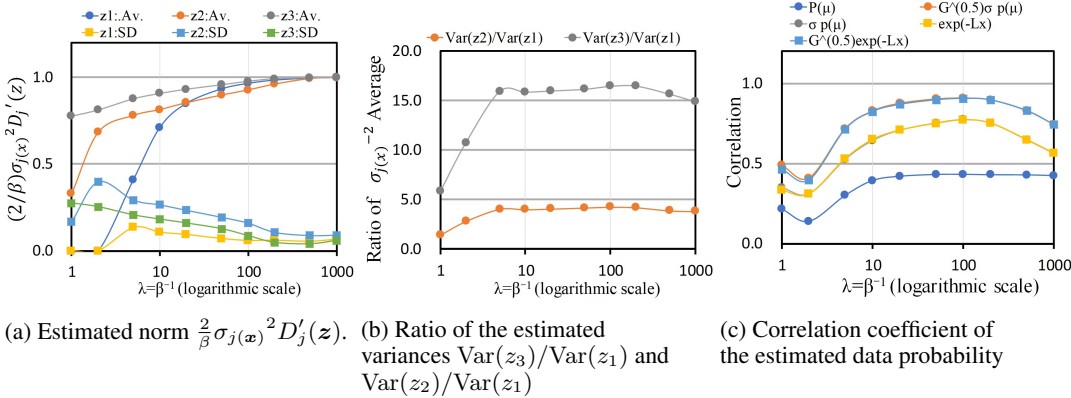

(a) Estimated norm $\frac{2}{\beta}\sigma_{j(\boldsymbol{x})}{}^2 D_j'(\boldsymbol{z})$.

(b) Ratio of the estimated variances $\mathrm{Var}(z_3)/\mathrm{Var}(z_1)$ and $\mathrm{Var}(z_2)/\mathrm{Var}(z_1)$

(c) Correlation coefficient of the estimated data probability

Figure 12: Property measurements of the Mix dataset using the downward-convex loss. $\lambda$ is changed from 1 to 1,000. $\mathrm{Var}(z_j)$ denotes the estimated variance, given by the average of $\sigma_{j(\boldsymbol{x})}^{-2}$.

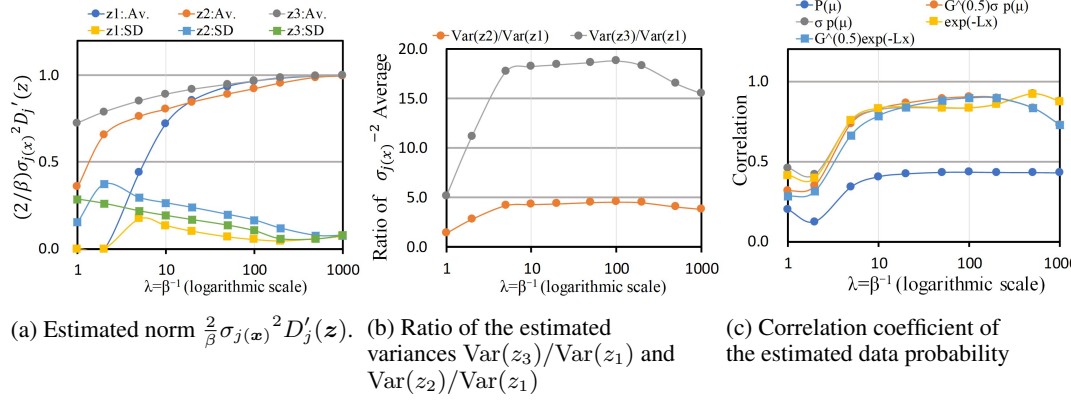

(a) Estimated norm $\frac{2}{\beta}\sigma_{j(\boldsymbol{x})}{}^2 D_j'(\boldsymbol{z})$.

(b) Ratio of the estimated variances $\mathrm{Var}(z_3)/\mathrm{Var}(z_1)$ and $\mathrm{Var}(z_2)/\mathrm{Var}(z_1)$

(c) Correlation coefficient of the estimated data probability

Figure 13: Property measurements of the Mix dataset using the upward-convex loss. $\lambda$ is changed from 1 to 1,000. $\mathrm{Var}(z_j)$ denotes the estimated variance, given by the average of $\sigma_{j(\boldsymbol{x})}^{-2}$.

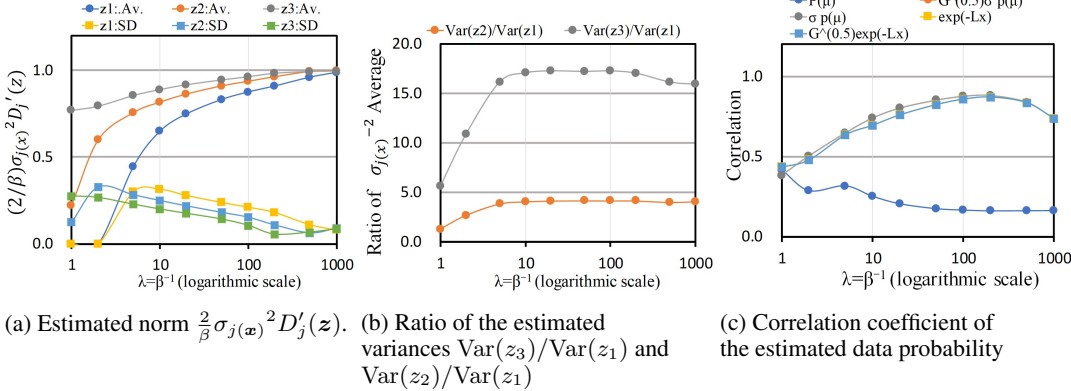

(a) Estimated norm $\frac{2}{\beta}\sigma_{j(\boldsymbol{x})}{}^2 D_j'(\boldsymbol{z})$.

(b) Ratio of the estimated variances $\mathrm{Var}(z_3)/\mathrm{Var}(z_1)$ and $\mathrm{Var}(z_2)/\mathrm{Var}(z_1)$

(c) Correlation coefficient of the estimated data probability

Figure 14: Property measurements of the Ramp dataset using the square error loss. $\lambda$ is changed from 1 to 1,000. $\mathrm{Var}(z_j)$ denotes the estimated variance, given by the average of $\sigma_{j(\boldsymbol{x})}^{-2}$.

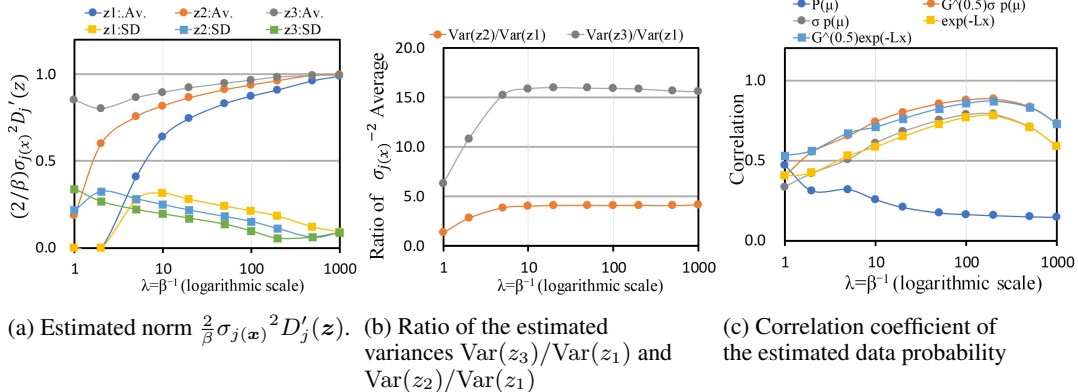

(a) Estimated norm $\frac{2}{\beta}\sigma_{j(\boldsymbol{x})}{}^2 D_j'(\boldsymbol{z})$.

(b) Ratio of the estimated variances $\mathrm{Var}(z_3)/\mathrm{Var}(z_1)$ and $\mathrm{Var}(z_2)/\mathrm{Var}(z_1)$

(c) Correlation coefficient of the estimated data probability

Figure 15: Property measurements of the Ramp dataset using the downward-convex loss. $\lambda$ is changed from 1 to 1,000. $\mathrm{Var}(z_j)$ denotes the estimated variance, given by the average of $\sigma_{j(\boldsymbol{x})}^{-2}$.

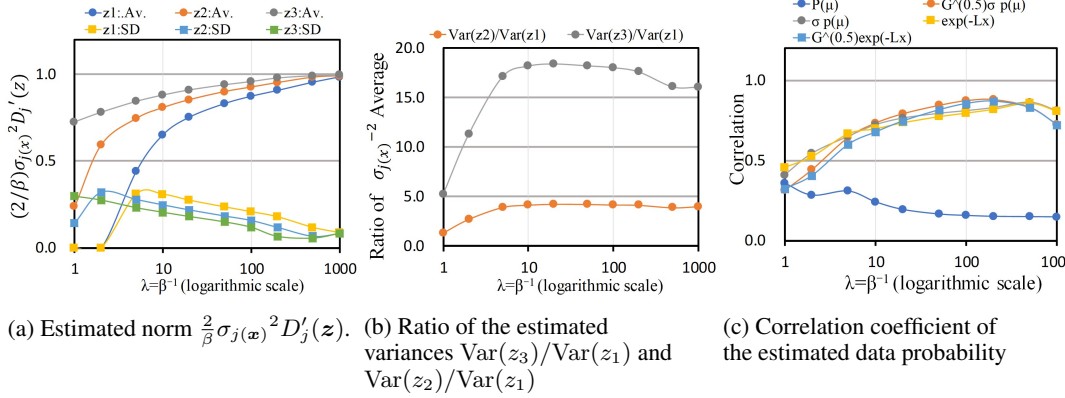

(a) Estimated norm $\frac{2}{\beta}\sigma_{j(\boldsymbol{x})}{}^2 D_j'(\boldsymbol{z})$.

(b) Ratio of the estimated variances $\mathrm{Var}(z_3)/\mathrm{Var}(z_1)$ and $\mathrm{Var}(z_2)/\mathrm{Var}(z_1)$

(c) Correlation coefficient of the estimated data probability

Figure 16: Property measurements of the Ramp dataset using the upward-convex loss. $\lambda$ is changed from 1 to 1,000. $\mathrm{Var}(z_j)$ denotes the estimated variance, given by the average of $\sigma_{j(\boldsymbol{x})}^{-2}$.

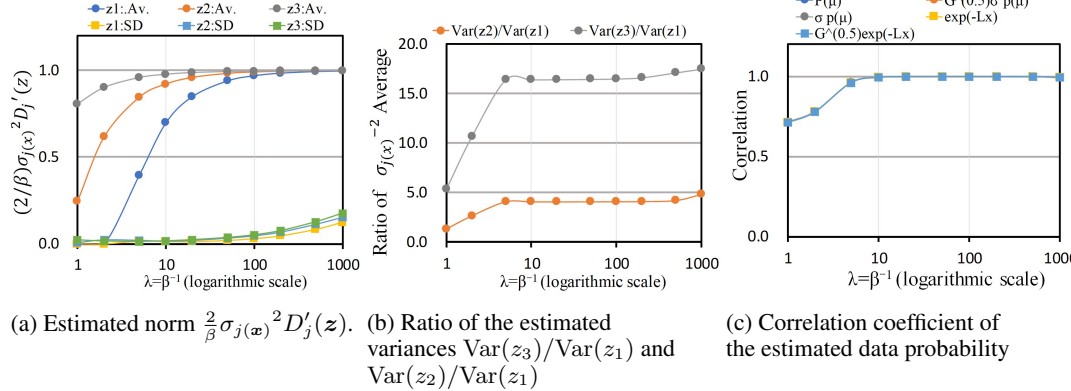

(a) Estimated norm $\frac{2}{\beta}\sigma_{j(\boldsymbol{x})}{}^2 D_j'(\boldsymbol{z})$.

(b) Ratio of the estimated variances $\mathrm{Var}(z_3)/\mathrm{Var}(z_1)$ and $\mathrm{Var}(z_2)/\mathrm{Var}(z_1)$

(c) Correlation coefficient of the estimated data probability

Figure 17: Property measurements of the Norm dataset using the square error loss. $\lambda$ is changed from 1 to 1,000. $\mathrm{Var}(z_j)$ denotes the estimated variance, given by the average of $\sigma_{j(\boldsymbol{x})}^{-2}$.

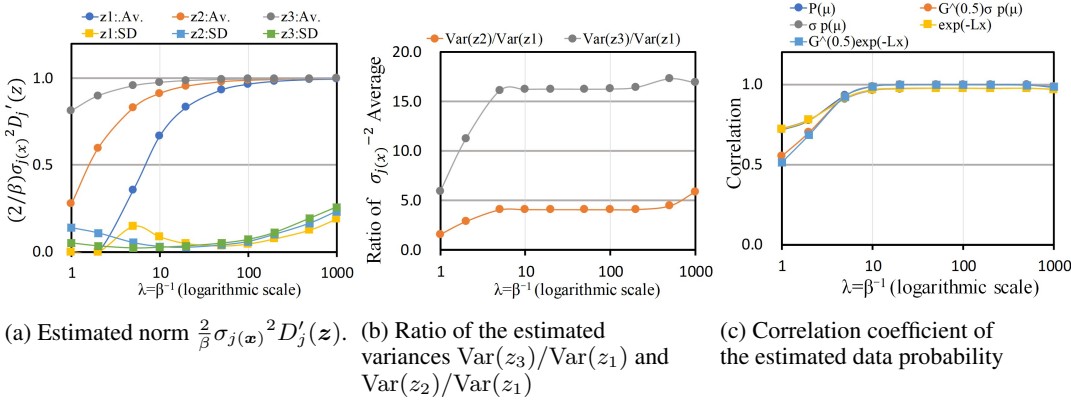

(a) Estimated norm $\frac{2}{\beta}\sigma_{j(\boldsymbol{x})}{}^2 D_j'(\boldsymbol{z})$.

(b) Ratio of the estimated variances $\mathrm{Var}(z_3)/\mathrm{Var}(z_1)$ and $\mathrm{Var}(z_2)/\mathrm{Var}(z_1)$

(c) Correlation coefficient of the estimated data probability

Figure 18: Property measurements of the Norm dataset using the downward-convex loss. $\lambda$ is changed from 1 to 1,000. $\mathrm{Var}(z_j)$ denotes the estimated variance, given by the average of $\sigma_{j(\boldsymbol{x})}^{-2}$.

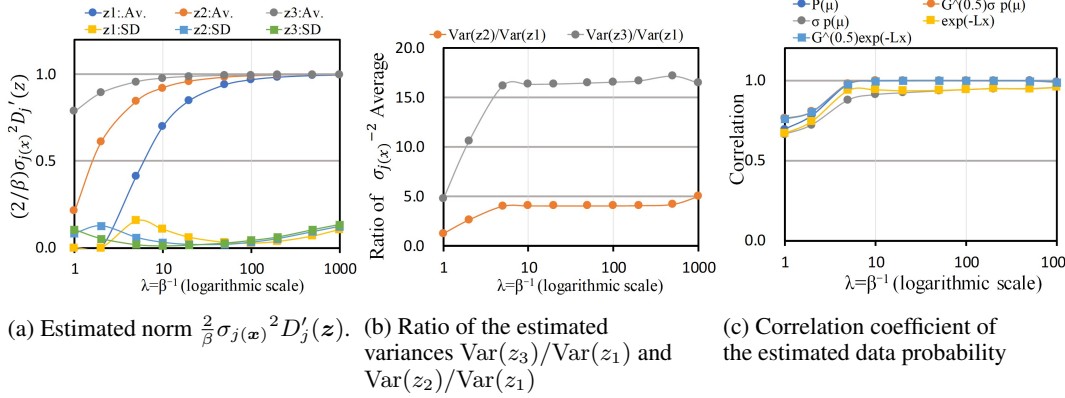

(a) Estimated norm $\frac{2}{\beta}\sigma_{j(\boldsymbol{x})}{}^2 D_j'(\boldsymbol{z})$.

(b) Ratio of the estimated variances $\mathrm{Var}(z_3)/\mathrm{Var}(z_1)$ and $\mathrm{Var}(z_2)/\mathrm{Var}(z_1)$

(c) Correlation coefficient of the estimated data probability

Figure 19: Property measurements of the Mix dataset using the upward-convex loss. $\lambda$ is changed from 1 to 1,000. $\mathrm{Var}(z_j)$ denotes the estimated variance, given by the average of $\sigma_{j(\boldsymbol{x})}^{-2}$.

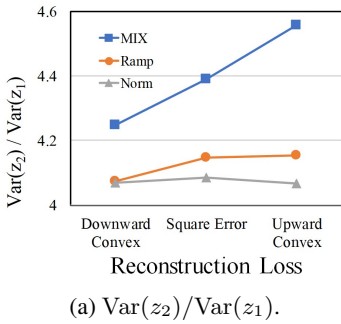

(a) $\mathrm{Var}(z_2)/\mathrm{Var}(z_1)$.

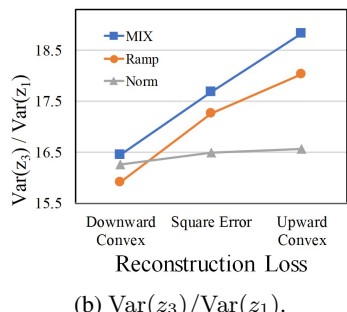

(b) $\mathrm{Var}(z_3)/\mathrm{Var}(z_1)$.

Figure 20: Ratio of the estimated variances $\mathrm{Var}(z_3)/\mathrm{Var}(z_1)$ and $\mathrm{Var}(z_2)/\mathrm{Var}(z_1)$ for the three datasets and three coding losses at $\lambda = 100$. $\mathrm{Var}(z_j)$ denotes the estimated variance, given by the average of $\sigma_{j(\boldsymbol{x})}^{-2}$.

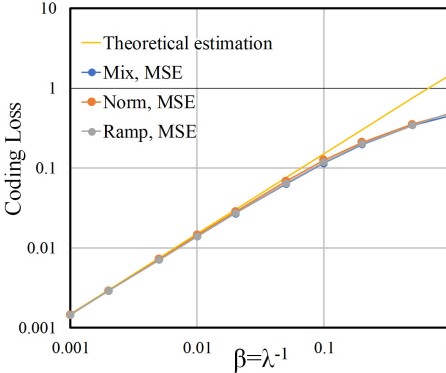

Figure 21: Dependency of Coding Loss on $\beta$ for Mix, Norm, and Ramp dataset using square loss.

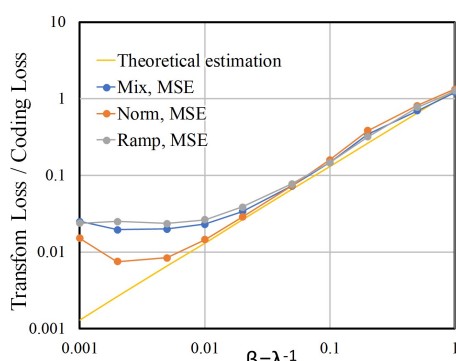

Figure 22: Dependency of Transform loss / Coding Loss Ratio on $\beta$ for Mix, Norm, and Ramp dataset using square loss.

# F   ADDITIONAL RESULTS IN CELEBA DATASET

## F.1   TRAVERSED OUTPUTS FOR ALL THE COMPONENT IN THE EXPERIMENTAL SECTION 4.2

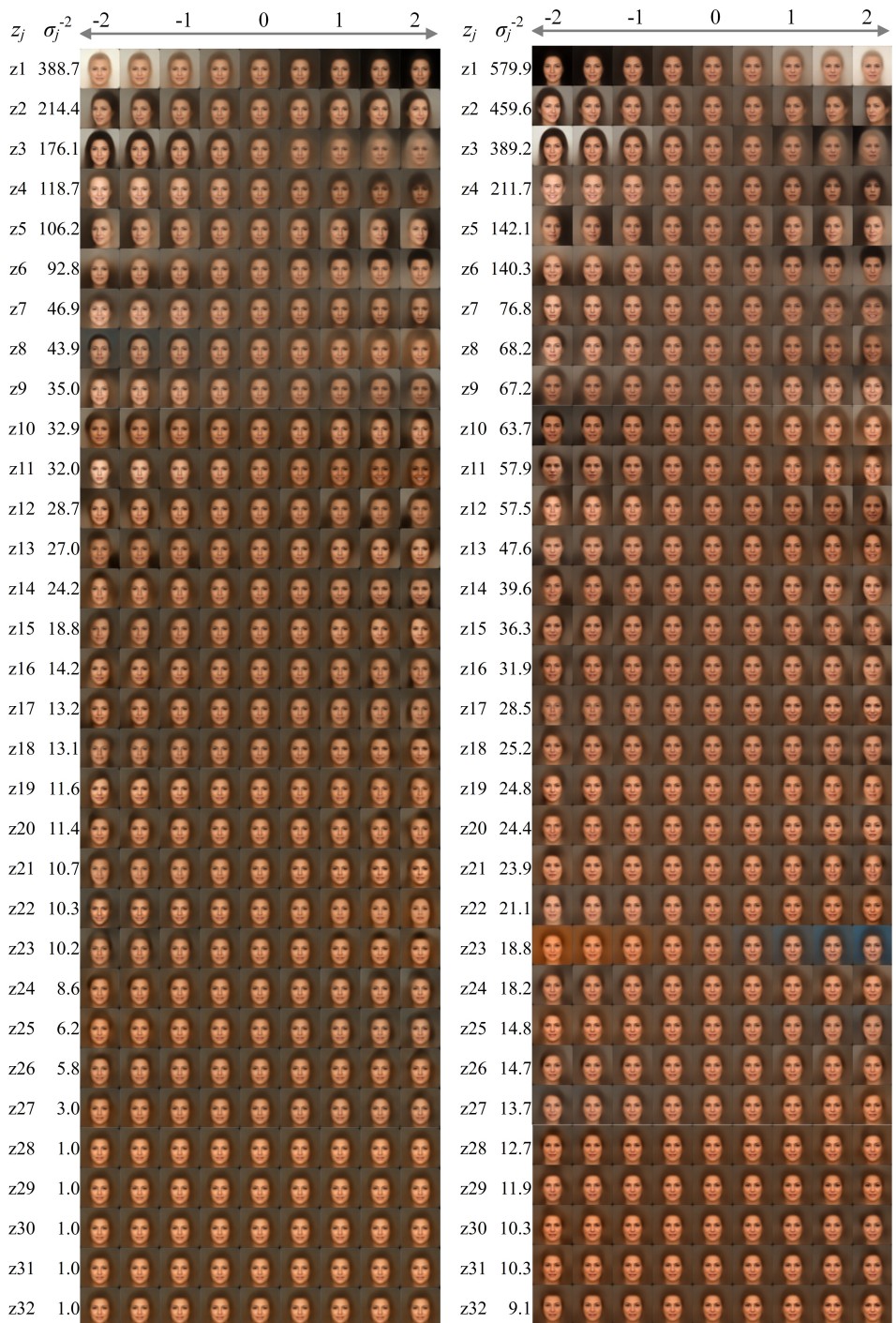

(a) Trained using the conventional loss form.    (b) Trained using the decomposed loss form.

Figure 23: Traversed outputs for all the component, changing $z_j$ from $-2$ to $2$. The latent variables $z_j$ are numbered in descending order by the estimated variance $\sigma_j^{-2}$ shown in Figures 4 and 5.

Figure 23 shows decoder outputs for all the components, where each latent variable is traversed from $-2$ to $2$. The estimated variance of each $y_j$, i.e., $\overline{\sigma_j^{-2}}$, is also shown in these figures. The latent variables $z_i$ are numbered in descending order by the estimated variances. Figure 23a is a result using the conventional loss form, i.e., $L_{\boldsymbol{x}} = D(\boldsymbol{x}, \hat{\boldsymbol{x}}) + \beta D_{\mathrm{KL}}(\cdot)$. The degrees of change seem to descend in accordance with the estimated variances. In the range where $j$ is 1 from 10, the degrees of changes are large. In the range $j > 10$, the degrees of changes becomes gradually smaller. Furthermore, almost no change is observed in the range $j > 27$. As shown in Figure 4, $D_{\mathrm{KL(j)}}(\cdot)$ is close to zero for $j > 27$, meaning no information. Thus, this result is clearly consistent with our theoretical analysis in Section

Figure 23b is a result using the decomposed loss form, i.e., $L_{\boldsymbol{x}} = D(\boldsymbol{x}, \check{\boldsymbol{x}}) + D(\check{\boldsymbol{x}}, \hat{\boldsymbol{x}}) + \beta D_{\mathrm{KL}}(\cdot)$. The degrees of change also seem to descend in accordance with the estimated variances. When looking at the detail, there are still minor changes even $j = 32$. As shown in Figure 5, KL divergences $D_{\mathrm{KL(j)}}(\cdot)$ for all the components are larger than zero. This implies all of the dimensional components have meaningful information. Therefore, we can see a minor change even $j = 32$. Thus, this result is also consistent with our theoretical analysis.

Another minor difference is sharpness. Although the quantitative comparison is difficult, the decoded images in Figure 23b seems somewhat sharper than those in Figure 23a. A possible reason for this minor difference is as follows. The transform loss $D(\boldsymbol{x}, \check{\boldsymbol{x}})$ serves to bring the decoded image of $\boldsymbol{\mu}_{(\boldsymbol{x})}$ closer to the input. In the conventional image coding, the orthonormal transform and its inverse transform are used for encoding and decoding, respectively. Therefore, the input and the decoded output are equivalent when not using quantization. If not so, the quality of the decoded image will suffer from the degradation. Considering this analogy, the use of decomposed loss might improve the decoded images for $\boldsymbol{\mu}_{(\boldsymbol{x})}$, encouraging the improvement of the orthonormality of the encoder/decoder in VAE.

## F.2 ADDITIONAL EXPERIMENTAL RESULT WITH OTHER CONDITION

In this Section, we provide the experimental results with other condition. We use essentially the same condition as described in Appendix D.2, except for the following conditions. The bottleneck size and $\lambda$ are set to 256 and 10000, respectively. The encoder network is composed of CNN(9, 9, 2, 64, GDN) - CNN(5, 5, 2, 64, GDN) - CNN(5, 5, 2, 64, GDN) - CNN(5, 5, 2, 64, GDN) - FC(1024, 2048, softplus) - FC(2048, 256, None)$\times 2$ (for $\boldsymbol{\mu}$ and $\boldsymbol{\sigma}$) in encoder. The decoder network is composed of FC(256, 2048, softplus) - FC(2048, 1024, softplus) - CNN(5, 5, 2, 64, IGDN) - CNN(5, 5, 2, 64, IGDN) - CNN(5, 5, 2, 64, IGDN)-CNN(9, 9, 2, 3, IGDN).

Figures 24a and 24b show the averages of $\sigma_{j(\boldsymbol{x})}^{-2}$ as well as the average and the standard deviation of $\frac{2}{\beta}\sigma_{j(\boldsymbol{x})}^2 D_j'(\boldsymbol{z})$ in the conventional loss form and the decomposed loss form, respectively. When using the conventional loss form, the mean of $\frac{2}{\beta}\sigma_{j(\boldsymbol{x})}^2 D_j'(\boldsymbol{z})$ is 1.25, which is closer to 1 than the

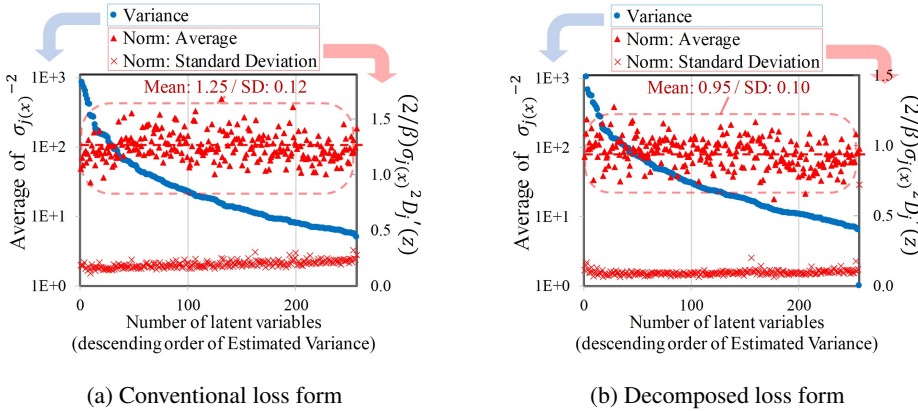

(a) Conventional loss form    (b) Decomposed loss form

Figure 24: Graph of $\sigma_{j(\boldsymbol{x})}^{-2}$ average and $\frac{2}{\beta}\sigma_{j(\boldsymbol{x})}^2 D_j'(\boldsymbol{z})$ in CelebA dataset. The bottleneck size and $\lambda$ are set to 256 and 10000, respectively.

mean 1.83 in Section 4.2. This suggests that the implicit transform is closer to the orthonormal. The possible reason is that a bigger reconstruction error is likely to cause the interference to RD-trade off and a slight violation of the theory, and it might be compensated with a larger lambda. When using the decomposed loss form, the mean of $\frac{2}{\beta}\sigma_{j(\boldsymbol{x})}{}^2 D'_j(\boldsymbol{z})$ is 0.95, meaning almost unit norm. These results also support that VAE provides the implicit orthonormal transform even if the lambda or bottleneck size is varied.

# G  ADDITIONAL EXPERIMENTAL RESULT WITH MNIST DATASET

In this Appendix, we provide the experimental result of Section 4.2 with MNIST dataset[3] consists of binary hand-written digits with a dimension of 768(=28 × 28). We use standard training split which includes 50,000 data points. For the reconstruction loss, we use the binary cross entropy loss (BCE) for the Bernoulli distribution. We averaged BCE by the number of pixels.

The encoder network is composed of FC(768, 1024, relu) - FC(1024, 1024, relu) - FC(1024, bottleneck size) in encoder. The decoder network is composed of FC(bottleneck size, 1024, relu) - FC(1024, 1024, relu) - FC(1024, 768, sigmoid). The batch size is 256 and the training iteration number is 50,000. In this section, results with two parameters, (bottleneck size=32, $\lambda$=2000) and (bottleneck size=64, $\lambda$=10000) are provided. Note that since we averaged BCE loss by the number of pixels, $\beta$ in the conventional $\beta$ VAE is derived by $768/\lambda$. Then, the model is optimized by Adam optimizer with the learning rate of 1e-3, using the conventional (not decomposed) loss form.

We use a PC with CPU Intel(R) Core(TM) i7-6850K CPU @ 3.60GHz, 12GB memory equipped with NVIDIA GeForce GTX 1080. The simulation time for each trial is about 10 minutes, including the statistics evaluation codes.

Figure 25 shows the averages of $\sigma_{j(\boldsymbol{x})}{}^{-2}$ as well as the average and the standard deviation of $\frac{2}{\beta}\sigma_{j(\boldsymbol{x})}{}^2 D'_j(\boldsymbol{z})$. In both conditions, the means of $\frac{2}{\beta}\sigma_{j(\boldsymbol{x})}{}^2 D'_j(\boldsymbol{z})$ averages are also close to 1 except in the dimensions where $\sigma_{j(\boldsymbol{x})}{}^{-2}$ is less than 10. These results suggest the theoretical property still holds when using the BCE loss. In the dimensions where $\sigma_{j(\boldsymbol{x})}{}^{-2}$ is less than 10, the $\frac{2}{\beta}\sigma_{j(\boldsymbol{x})}{}^2 D'_j(\boldsymbol{z})$ is somewhat lower than 1. The possible reason is that $D_{\mathrm{KL(j)}}(\cdot)$ in such dimension is 0 for some inputs and is larger than 0 in other inputs. The understanding of the transition region needs further study.

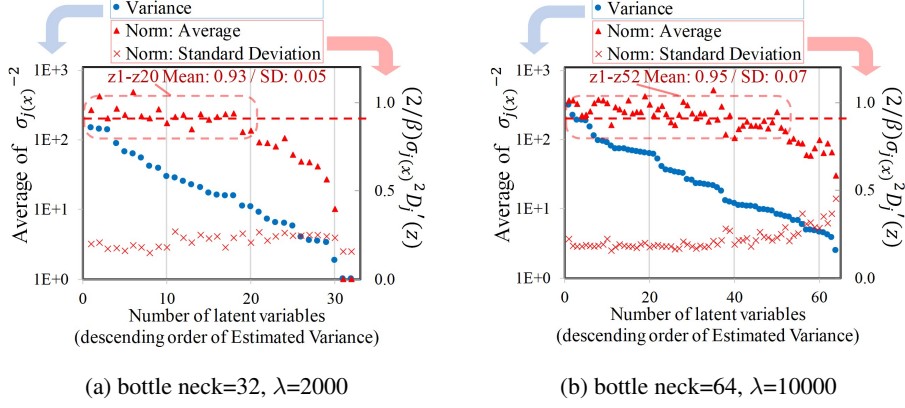

(a) bottle neck=32, $\lambda$=2000          (b) bottle neck=64, $\lambda$=10000

Figure 25: Graph of $\sigma_{j(\boldsymbol{x})}{}^{-2}$ average and $\frac{2}{\beta}\sigma_{j(\boldsymbol{x})}{}^2 D'_j(\boldsymbol{z})$ in MNIST dataset.

---

[3]http://yann.lecun.com/exdb/mnist/

# H   DERIVATION/EXPLANATION IN RDO-RELATED EQUATION EXPANSIONS

## H.1   APPROXIMATION OF DISTORTION IN UNIFORM QUANTIZATION

Let $T$ be a quantization step. Quantized values $\hat{z}_j$ is derived as $k\,T$, where $k = \mathrm{round}(z_j/T)$. Then $d$, the distortion per channel, is approximated by

$$
\begin{aligned}
d &= \sum_k \int_{(k-1/2)T}^{(k+1/2)T} p(z_j)(z_j - k\,T)^2 \; \mathrm{d}z_j \\
&\simeq \sum_k T\,p(k\,T) \int_{(k-1/2)T}^{(k+1/2)T} \frac{1}{T}(z_j - k\,T)^2 \; \mathrm{d}z_j \\
&= \frac{T^2}{12} \sum_k T\,p(k\,T) \\
&\simeq \frac{T^2}{12}.
\end{aligned}
\tag{67}
$$

Here, $\sum_k T\,p(k\,T) \simeq \int_{-\infty}^{\infty} p(z_j)\mathrm{d}z_j = 1$ is used. The distortion for the given quantized value is also estimated as $T^2/12$, because this value is approximated by $\int_{(k-1/2)T}^{(k+1/2)T} \frac{1}{T}(z_j - k\,T)^2 \; \mathrm{d}z_j$.

## H.2   APPROXIMATION OF RECONSTRUCTION LOSS AS A QUADRATIC FORM.

In this appendix, the approximations of the reconstruction losses as a quadratic form ${}^t\delta\boldsymbol{x}\,\boldsymbol{G_x}\delta\boldsymbol{x}+C_{\boldsymbol{x}}$ are explained for the sum of square error (SSE), binary cross entropy (BCE) and Structural Similarity (SSIM). Here, we have borrowed the derivation of BCE and SSIM from Kato et al. (2020), and add some explanation and clarification to them for convenience. We also describe the log-likelihood of the Gaussian distribution.

Let $\hat{\boldsymbol{x}}$ and $\hat{x}_i$ be decoded sample $\mathrm{Dec}_\theta(\boldsymbol{z})$ and its $i$-th dimensional component respectively. $\delta\boldsymbol{x}$ and $\delta x_i$ denote $\boldsymbol{x} - \hat{\boldsymbol{x}}$ and $x_i - \hat{x}_i$, respectively. It is also assumed that $\delta\boldsymbol{x}$ and $\delta x_i$ are infinitesimal. The details of the approximations are described as follows.

**Sum square error:**
In the case of sum square error, $\boldsymbol{G_x}$ is equal to $\boldsymbol{I}_m$. This can be derived as:

$$
\sum_{i=1}^m (x_i - \hat{x}_i)^2 = \sum_{i=1}^m \delta x_i^2 = {}^t\delta\boldsymbol{x}\boldsymbol{I}_m\delta\boldsymbol{x}.
\tag{68}
$$

**Binary cross entropy:**
Binary cross entropy is a log likelihood of the Bernoulli distribution. The Bernoulli distribution is described as:

$$
p_\theta(\boldsymbol{x}|\boldsymbol{z}) = \prod_{i=1}^m \hat{x}_i^{\,x_i}\,(1 - \hat{x}_i)^{(1-x_i)}
\tag{69}
$$

Then, the binary cross-entropy (BCE) can be expanded as:

$$
\begin{aligned}
-\log p_\theta(\boldsymbol{x}|\boldsymbol{z}) &= -\log \prod_{i=1}^m \hat{x}_i^{\,x_i}\,(1 - \hat{x}_i)^{(1-x_i)} \\
&= \sum_{i=1}^m (-x_i \log \hat{x}_i - (1 - x_i)\log(1 - \hat{x}_i)) \\
&= \sum_i (-x_i \log(x_i + \delta x_i) - (1 - x_i)\log(1 - x_i - \delta x_i)) \\
&= \sum_i \left(-x_i \log\left(1 + \frac{\delta x_i}{x_i}\right) - (1 - x_i)\log\left(1 - \frac{\delta x_i}{1 - x_i}\right)\right) \\
&\quad + \sum_i (-x_i \log(x_i) - (1 - x_i)\log(1 - x_i)).
\end{aligned}
\tag{70}
$$

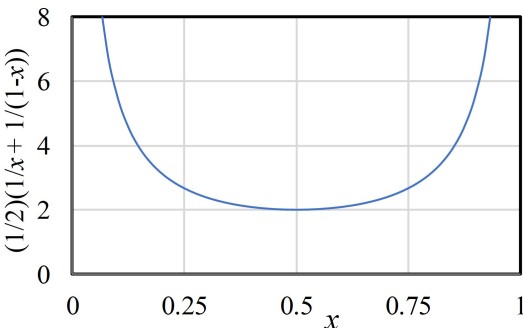

Figure 26: Graph of $\frac{1}{2}\left(\frac{1}{x} + \frac{1}{1-x}\right)$ in the BCE approximation.

Here, the second term of the last equation is a constant $C_{\boldsymbol{x}}$ depending on $\boldsymbol{x}$. Using $\log(1 + x) = x - x^2/2 + O(x^3)$, the first term of the last equation is further expanded as follows:

$$\sum_i \left(-x_i\left(\frac{\delta x_i}{x_i} - \frac{\delta x_i{}^2}{2x_i{}^2}\right) - (1 - x_i)\left(-\frac{\delta x_i}{1 - x_i} - \frac{\delta x_i{}^2}{2\left(1 - x_i\right)^2}\right) + O\left(\delta x_i{}^3\right)\right)$$

$$= \sum_i \left(\frac{1}{2}\left(\frac{1}{x_i} + \frac{1}{1 - x_i}\right)\delta x_i{}^2 + O\left(\delta x_i{}^3\right)\right). \tag{71}$$

As a result, a metric tensor $\boldsymbol{G_x}$ can be approximated as the following positive definite Hermitian matrix:

$$\boldsymbol{G_x} = \begin{pmatrix} \frac{1}{2}\left(\frac{1}{x_1} + \frac{1}{1-x_1}\right) & 0 & \cdots \\ 0 & \frac{1}{2}\left(\frac{1}{x_2} + \frac{1}{1-x_2}\right) & \cdots \\ \vdots & \vdots & \ddots \end{pmatrix}. \tag{72}$$

Here, the loss function in each dimension $\frac{1}{2}\left(\frac{1}{x_1} + \frac{1}{1-x_1}\right)$ is a downward-convex function as shown in Figure 26.

**Structural similarity (SSIM):**
Structural similarity (SSIM) (Wang et al., 2001) is widely used for picture quality metric, which is close to human subjective quality. Let SSIM be a SSIM value between two pictures. The range of the SSIM value is between 0 and 1. The higher the value, the better the quality. In this appendix, we also show that $(1 - \text{SSIM})$ can be approximated to a quadratic form such as ${}^t\delta\boldsymbol{x}\ \boldsymbol{G_x}\delta\boldsymbol{x}$.

$\text{SSIM}_{N\times N(h,v)}(\boldsymbol{x}, \boldsymbol{y})$ denotes a SSIM value between $N \times N$ windows in pictures $X$ and $Y$, where $\boldsymbol{x} \in \mathbb{R}^{N^2}$ and $\boldsymbol{y} \in \mathbb{R}^{N^2}$ denote $N \times N$ pixels cropped from the top-left coordinate $(h, v)$ in the images $X$ and $Y$, respectively. Let $\mu_{\boldsymbol{x}}$, $\mu_{\boldsymbol{y}}$ be the averages of all dimensional components in $\boldsymbol{x}$, $\boldsymbol{y}$, and $\sigma_{\boldsymbol{x}}$ , $\sigma_{\boldsymbol{y}}$ be the variances of all dimensional components in $\boldsymbol{x}$, $\boldsymbol{y}$ in the $N \times N$ windows, respectively. Then, $\text{SSIM}_{N\times N(h,v)}(\boldsymbol{x}, \boldsymbol{y})$ is derived as

$$\text{SSIM}_{N\times N(h,v)}(\boldsymbol{x}, \boldsymbol{y}) = \frac{2\mu_{\boldsymbol{x}}\mu_{\boldsymbol{y}}}{\mu_{\boldsymbol{x}}{}^2 + \mu_{\boldsymbol{y}}{}^2} \cdot \frac{2\sigma_{\boldsymbol{xy}}}{\sigma_{\boldsymbol{x}}{}^2 + \sigma_{\boldsymbol{y}}{}^2}. \tag{73}$$

In order to calculate a SSIM value for a picture, the window is shifted in a whole picture and all of SSIM values are averaged. Therefore, if $\left(1 - \text{SSIM}_{N\times N(h,v)}(\boldsymbol{x}, \boldsymbol{y})\right)$ is expressed as a quadratic form ${}^t\delta\boldsymbol{x}\ \boldsymbol{G}_{(h,v)\boldsymbol{x}}\ \delta\boldsymbol{x}$, $(1 - \text{SSIM})$ can be also expressed in quadratic form ${}^t\delta\boldsymbol{x}\ \boldsymbol{G_x}\delta\boldsymbol{x}$.

Let $\delta\boldsymbol{x}$ be a minute displacement of $\boldsymbol{x}$. $\mu_{\delta\boldsymbol{x}}$ and $\sigma_{\delta\boldsymbol{x}}{}^2$ denote an average and variance of all dimensional components in $\delta\boldsymbol{x}$, respectively. Then, SSIM between $\boldsymbol{x}$ and $\boldsymbol{x} + \delta\boldsymbol{x}$ can be approximated as:

$$\text{SSIM}_{N\times N(h,v)}(\boldsymbol{x}, \boldsymbol{x} + \delta\boldsymbol{x}) \simeq 1 - \frac{\mu_{\delta\boldsymbol{x}}{}^2}{2\mu_{\boldsymbol{x}}{}^2} - \frac{\sigma_{\delta\boldsymbol{x}}{}^2}{2\sigma_{\boldsymbol{x}}{}^2} + O\left(\left(|\delta\boldsymbol{x}|/|\boldsymbol{x}|\right)^3\right). \tag{74}$$

Then $\mu_{\delta\boldsymbol{x}}{}^2$ and $\sigma_{\delta\boldsymbol{x}}{}^2$ can be expressed as

$$\mu_{\delta\boldsymbol{x}}{}^2 = {}^t\delta\boldsymbol{x}\,\boldsymbol{M}\delta\boldsymbol{x}, \quad \text{where} \quad \boldsymbol{M} = \frac{1}{N^2}\begin{pmatrix} 1 & 1 & \cdots & 1 \\ 1 & 1 & \cdots & 1 \\ \vdots & \vdots & \ddots & \vdots \\ 1 & 1 & \cdots & 1 \end{pmatrix}, \tag{75}$$

and

$$\sigma_{\delta\boldsymbol{x}}{}^2 = {}^t\delta\boldsymbol{x}\,\boldsymbol{V}\delta\boldsymbol{x}, \quad \text{where} \quad \boldsymbol{V} = \frac{1}{N}\boldsymbol{I}_N - M, \tag{76}$$

respectively. As a result, $\left(1 - \mathrm{SSIM}_{N\times N(h,v)}(\boldsymbol{x}, \boldsymbol{x} + \delta\boldsymbol{x})\right)$ can be expressed in the following quadratic form as:

$$1 - \mathrm{SSIM}_{N\times N(h,v)}(\boldsymbol{x}, \boldsymbol{x} + \delta\boldsymbol{x}) \simeq {}^t\delta\boldsymbol{x}\,\boldsymbol{G}_{(h,v)\boldsymbol{x}}\delta\boldsymbol{x}, \quad \text{where} \quad \boldsymbol{G}_{(h,v)\boldsymbol{x}} = \left(\frac{1}{2\mu_x{}^2}\boldsymbol{M} + \frac{1}{2\sigma_x{}^2}\boldsymbol{V}\right). \tag{77}$$

It is noted that $\boldsymbol{M}$ is a positive definite Hermitian matrix and $\boldsymbol{V}$ is a positive semidefinite Hermitian matrix. Therefore, $\boldsymbol{G}_{(h,v)\boldsymbol{x}}$ is a positive definite Hermitian matrix. As a result, $(1 - \mathrm{SSIM})$ can be also expressed in quadratic form ${}^t\delta\boldsymbol{x}\,\boldsymbol{G}_{\boldsymbol{x}}\delta\boldsymbol{x}$, where $\boldsymbol{G}_{\boldsymbol{x}}$ is a positive definite Hermitian matrix.

**Log-likelihood of Gaussian distribution:**
Gaussian distribution is described as:

$$p_\theta(\boldsymbol{x}|\boldsymbol{z}) = \prod_{i=1}^{m} \frac{1}{\sqrt{2\pi\sigma^2}} e^{-(x_i - \hat{x}_i)^2/2\sigma^2} = \prod_{i=1}^{m} \frac{1}{\sqrt{2\pi\sigma^2}} e^{-\delta x_i{}^2/2\sigma^2}, \tag{78}$$

where $\sigma^2$ is a variance as a hyper parameter. Then, the log-likelihood of the Gaussian distribution is denoted as:

$$-\log p_\theta(\boldsymbol{x}|\boldsymbol{z}) = -\log \prod_{i=1}^{m} \frac{1}{\sqrt{2\pi\sigma^2}} e^{-\delta x_i^2/2\sigma^2} = \frac{1}{2\sigma^2}\sum_{i=1}^{m} \delta x_i^2 + \frac{m}{2}\log(2\pi\sigma^2). \tag{79}$$

The first term can be rewritten as $(1/2\sigma^2)\,{}^t\delta\boldsymbol{x}\boldsymbol{I}_m\delta\boldsymbol{x}$. Thus, $\boldsymbol{G}_{\boldsymbol{x}} = (1/2\sigma^2)\,\boldsymbol{I}_m$ holds. $C_{\boldsymbol{x}}$ is derived as the second term of the last equation in Eq.78.

### H.3 DETAILED EXPLANATION OF KL DIVERGENCE AS A RATE OF ENTROPY CODING.

This appendix explains the detail how KL divergence can be interpreted as a rate in the transform coding. In the transform coding, input data is transformed by an orthonormal transform. Then, the transformed data is quantized, and an entropy code is assigned to the quantized symbol, such that the length of the entropy code is equivalent to the logarithm of the estimated symbol probability.

It is generally intractable to derive the rate and distortion of individual symbols in the ideal information coding. Thus, we first discuss the case of uniform quantization. Let $P_{z_j}$ and $R_{z_j}$ be the probability and rate in the uniform quantization coding of $z_j \sim \mathcal{N}(z_j; 0, 1)$. Here, $\mu_{j(\boldsymbol{x})}$ and $\sigma_{j(\boldsymbol{x})}{}^2$ are regarded as a quantized value and a coding noise after the uniform quantization, respectively. Let $T$ be a quantization step size. The coding noise after quantization is $T^2/12$ for the quantization step size $T$, as explained in Appendix H.1. Thus, $T$ is derived as $T = 2\sqrt{3}\sigma_{j(\boldsymbol{x})}$ from $\sigma_{j(\boldsymbol{x})}{}^2 = T^2/12$. We also assume $\sigma_{j(\boldsymbol{x})}{}^2 \ll 1$. As shown in Fig.27a, $P_{z_j}$ is denoted by $\int_{\mu_{j(\boldsymbol{x})}-T/2}^{\mu_{j(\boldsymbol{x})}+T/2} p(z_j)\mathrm{d}z_j$ where $p(z_j)$ is $\mathcal{N}(z_j; 0, 1)$. Using Simpson's numerical integration method and $e^x = 1 + x + O(x^2)$ expansion, $P_{z_j}$ is approximated as:

$$\begin{aligned} P_{z_j} &\simeq \frac{T}{6}\left(p(\mu_{j(\boldsymbol{x})} - \tfrac{T}{2}) + 4p(\mu_{j(\boldsymbol{x})}) + p(\mu_{j(\boldsymbol{x})} + \tfrac{T}{2})\right) \\ &= \frac{Tp(\mu_{j(\boldsymbol{x})})}{6}\left(4 + e^{\frac{4\mu_{j(\boldsymbol{x})}T - T^2}{8}} + e^{\frac{-4\mu_{j(\boldsymbol{x})}T - T^2}{8}}\right) \\ &\simeq Tp\left(\mu_{j(\boldsymbol{x})}\right)\left(1 - T^2/24\right) \\ &= \sqrt{\frac{6}{\pi}}\sigma_{j(\boldsymbol{x})}\,e^{-(\mu_{j(\boldsymbol{x})}{}^2)/2}\left(1 - \frac{\sigma_{j(\boldsymbol{x})}{}^2}{2}\right). \end{aligned} \tag{80}$$

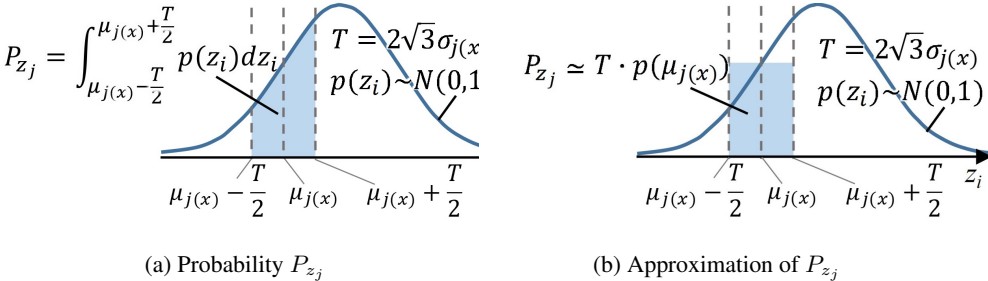

(a) Probability $P_{z_j}$  (b) Approximation of $P_{z_j}$

Figure 27: Probability for a symbol with mean $\mu$ and noise $\sigma^2$

Using $\log(1+x) = x + O(x^2)$ expansion, $R_{\mu\sigma}$ is derived as:

$$R_{z_j} = -\log P_{z_j} \simeq \frac{1}{2}\left(\mu_{j(\boldsymbol{x})}{}^2 + \sigma_{j(\boldsymbol{x})}{}^2 - \log \sigma_{j(\boldsymbol{x})}{}^2 - \log\frac{6}{\pi}\right) = D_{\mathrm{KL}j(\boldsymbol{x})}(\cdot) + \frac{1}{2}\log\frac{\pi e}{6}. \quad (81)$$

When $R_{z_j}$ and $D_{\mathrm{KL}j(\boldsymbol{x})}(\cdot)$ in Eq. 2 are compared, both equations are equivalent except a small constant difference $\frac{1}{2}\log(\pi e/6) \simeq 0.176$ for each dimension. As a result, KL divergence for $j$-th dimension is equivalent to the rate for the uniform quantization coding, allowing a small constant difference.

To make theoretical analysis easier, we use the simpler approximation as $P_{z_j} = T\ p(\mu_{j(\boldsymbol{x})}) = 2\sqrt{3}\sigma_{j(\boldsymbol{x})}\ p(\mu_{j(\boldsymbol{x})})$ instead of Eq.80, as shown in Fig.27b. Then, $R_{z_j}$ is derived as:

$$R_{z_j} = -\log(2\sqrt{3}\ \sigma_{j(\boldsymbol{x})}\ p(\mu_{j(\boldsymbol{x})})) = \text{Eq. 6} + \frac{1}{2}\log\frac{\pi e}{6}. \quad (82)$$

This equation also means that the approximation of KL divergence in Eq. 6 is equivalent to the rate in the uniform quantization coding with $P_{z_j} = 2\sqrt{3}\sigma_{j(\boldsymbol{x})}\ p(\mu_{j(\boldsymbol{x})})$ approximation, allowing the same small constant difference as in Eq. 81. It is noted that the approximation $P_{z_j} = 2\sqrt{3}\sigma_{j(\boldsymbol{x})}\ p(\mu_{j(\boldsymbol{x})})$ in Figure 27b can be applied to any kinds of prior PDFs because there is no explicit assumption for the prior PDF. This implies that the theoretical discussion after Eq. 6 in the main text will hold in arbitrary prior PDFs.

Finally, the meaning of the small constant difference $\frac{1}{2}\log\frac{\pi e}{6}$ in Eqs. 81 and 82 is shown. Pearlman & Said (2011) explains that the difference of the rate between the ideal information coding and uniform quantization is $\frac{1}{2}\log\frac{\pi e}{6}$. This is caused by the entropy difference of the noise distributions. In the ideal case, the noise distribution is known as a Gaussian. In the case the noise variance is $\sigma^2$, the entropy of the Gaussian noise is $\frac{1}{2}\log(\sigma^2 2\pi e)$. For the uniform quantization with a uniform noise distribution, the entropy is $\frac{1}{2}\log(\sigma^2 12)$. As a result, the difference is just $\frac{1}{2}\log\frac{\pi e}{6}$. Because the rate estimation in this appendix uses a uniform quantization, the small offset $\frac{1}{2}\log\frac{\pi e}{6}$ can be regarded as a difference between the ideal information coding and the uniform quantization. As a result, KL divergence in Eq. 2 and Eq. 6 can be regarded as a rate in the ideal informaton coding for the symbol with the mean $\mu_{j(\boldsymbol{x})}$ and variance $\sigma_{j(\boldsymbol{x})}{}^2$.

