# OpenReview forum: "Quantitative Understanding of VAE as a Non-linearly Scaled Isometric Embedding"
_ICLR.cc/2021/Conference — Reject_

### Official Review · AnonReviewer4 · 2020-10-28
**A quantitive study of the VAE, but with low clarity.**

**Rating:** 4
**Confidence:** 3

**Review:**

Summary: The paper proposes a methodology to understand quantitively the VAE model, based on the Rate-distortion theory.

Comments: I think that the writing of the paper is rather confusing. Unfortunately, I am not able to judge the proposed idea (which might be interesting) and provide a reasonable review, because I could not understand it after trying to read the paper couple of times. In general, I find the clarity and coherence of the paper lacking. In my opinion, the paper should be improved significantly along these lines, such that to be accessible from the reader.
​
Some examples of the problems I think that should be fixed:
​
- The text is super "wordy". There is too much compact information, but with low clarity. In general, I could not understand the main story of the paper.
- Many terms are used without a definition e.g. what is the definition of isometric embedding? what is the definition of rate-distortion?
- In general, the mathematical notation that is used is a bit unclear.
- There are many places where the authors say: "We (will) show later ...". I believe that this causes a confusion. I think is better to avoid this tactic and directly show the result in place.
- How the VAE model is perceived? In particular, I cannot understand what it means to "map the VAE"?
- In my opinion the Figure 1 is hard to understand.
- There is a lot of context and connections to related works, which the authors presume is already known from the reader. However, I think that this is not the case for a general reader.
​
Overall: Unfortunately, I am not able to provide a constructive feedback. The main idea of the paper might be interesting, but the writing style does not help. Therefore, I believe that the paper needs to be improved significantly, such that to be accessible.

===== After rebuttal =====

I appreciate the fact that the authors update their manuscript, taking into account some of our comments (+1 for this). However, I still feel that the content is quite hard to follow. In general, I believe that the motivation is still unclear and the clarity low. Even Fig. 1 (that potentially should help the understanding) is so complicated without any explanation in the caption. Maybe the technical content is interesting, but I think that the paper is not accessible.

---

> ### Author Response · Authors · 2020-11-22
> **Thank you very much for your careful comments**
>
> Thank you very much for your careful review.
>
> Although we agree that our description is "wordy," please understand this is a trade-off with self-contained explanations. We carefully describe our equations and proofs in a self-contained form such that it will be able to follow the equations without the knowledge of rate-distortion theory, Wiener filter, RaDOGAGA, and isometric embedding. We revised some parts for easier understanding. We hope this will help you understand our paper deeper and clearer.
>
> We agree that there are many places where the authors say: "We (will) show later ...". We revised all of them.
>
> Regarding the description "map the VAE", we revised such that "mapping VAE latent space to an implicit isometric space like RaDOGAGA on a variable-by-variable basis." We hope this description will be clearer.
>
> We hope our explanations and revision will help you to understand our paper clearer and deeper.

---

### Official Review · AnonReviewer3 · 2020-10-28
**Potentially interesting, but difficult to understand**

**Rating:** 5
**Confidence:** 2

**Review:**

This paper analyses the theoretical properties of the VAE, showing that under some assumptions the VAE can be interpreted as an implicit isometric embedding of the data manifold. The authors show that with this analogy one can make inferences about the data distribution and the latent dimensions from the components of the VAE.

It seems to me on the whole that the theoretical analysis and contribution is correct and solid, and the work does give a valuable insight into the theoretical properties of the VAE as interpreted as a isometric embedding of the data space into the latent space. The empirical results given demonstrate that the quantitative measures the authors describe give a way to understand the importance of the latents and the properties of the learned approximate data distribution. These are things that I believe the community would find useful.

My main negative impression of this paper is that it is hard to read. The paper is written in a way which is hard to understand. It is information-dense, with a lack of intuitive explanation for many of the statements. For example, the introduction of the rate-distortion theory is not motivated at all. Clearly, the reason it has been introduced is to draw the parallel to lossy data compression via mapping to a low dimensional latent space. This parallel allows one to use knowledge from the compression literature to aid our understanding of these autoencoder-like models. However, this is not mentioned anywhere in the text, which makes it difficult for the reader to follow the narrative of the paper. Note that the rate-distortion is just one example. I will say that I am not very familiar with the literature on isometric embeddings, which has made it harder for me to understand.

One other negative point I would raise is that the paper is similar in nature to the RaDOGAGA work by Kato et al. Their method also establishes that a VAE-like architecture can isometrically embed the data into the latent space, which allows them to draw strong inferences about the data distribution, given the isometric properties. At a high level, this is very similar to the contribution of this paper. There are certainly differences, primarily in the theoretical analysis given to the VAE here, and the focus on examining the contributions of each latent dimension through the variances, in a PCA-like fashion. Still, I think perhaps the novelty is lacking.

Overall, I think the paper should be rejected, but I do not feel strongly given that I am not an expert in the subject matter that much of the paper pertains to. If the differences between this work and RaDOGAGA were made, and the paper itself was made more clear then I would be willing to upgrade my score.

---

> ### Author Response · Authors · 2020-11-22
> **Thank you very much for your careful comments**
>
> Thank you very much for your careful review.
>
> As you pointed out, RaDOGAGA is our start-point. However, RaDOGAGA did not clarify the behavior of latent space and ELBO in VAE after optimization quantitatively, which has been discussed but not well clarified.
> We show that VAE objective becomes a log-likelihood of the input.  That is, VAE has already achieved the information bottleneck (Tishby et al. 1999) as it is.  As a result, the theoretical analysis "ELBO is broken" in  Alemi et al. (2018) is incorrect.  We also show what beta-VAE does is only to scale the metrics by a factor of beta from the original VAE.
>
> In the newly added section 3.4, we described such quantitative behaviors of VAE.  These explanations were mainly described in the previous Appendix A, and we moved them to section 3.4 in the main text to clarify the improvement from RaDOGAGA. As a result, we can thoroughly clarify the behavior of VAE and further spur the information-theoretic generative model.
>
> We agree that our description is lacking an overview of the motivation.  In the introduction, we added motivation and some explanation as below:
>  - Clarify what has not been discussed in prior works.
>  - Clarify the motivation why Rate-distortion theory is introduced.
>  - Brief explanation of isometric embedding.
>
> Although we agree that our description is "Information-dense", please understand this is a trade-off with self-contained explanations.
> We carefully describe our equations and proofs are self-contained such that it will be able to follow the equations without the knowledge of rate-distortion theory, Wiener filter, RaDOGAGA, and isometric embedding. We revised some parts for easier understanding.  We hope this will help you understand our paper deeper and clearer.
>
> We would appreciate it if you consider our clarification.

---

### Official Review · AnonReviewer1 · 2020-10-28
**Interesting interpretation of a VAE; but the paper might not be self-contained.**

**Rating:** 5
**Confidence:** 1

**Review:**

Based mainly on the rate-distortion theory, the authors propose to interpret VAE as a non-linearly scaled isometric embedding. That interpretation brings several advantages, for example, enabling the estimation of the data probabilities in the input space.

Overall, the paper is well written. However, I find it challenging to fully understand the main contributions of the paper, even though I use VAE a lot.

The current manuscript might not be self-contained. It relies heavily on the prior work RaDOGAGA (Kato et al., 2020), with which I am not familiar. I would suggest the authors elaborate on that.

The main interpretation/discussions are proposed for the classical VAE with a (conditional) isometric Gaussian variational arm, whose model capacity may be limited in practice. So, is it possible that the proposed interpretation/discussions also generalize to other VAE variants (like one exploiting a flow on top of the variational arm)?

In Section 3.2, many approximations are employed; what’s the influence of those approximations? Discussions are necessary.

In Eq (8) as well as several other equations, implicit assumptions have been made, that is, the Jacobian determinant is always positive; however, those assumptions might not be true in practice. How to address related concerns in practice?

---

> ### Author Response · Authors · 2020-11-22
> **Thank you very much for your careful comments**
>
> Thank you very much for your valuable and careful review.
>
> Thank you very much for pointing out the difference between VAE and RaDOGAGA is unclear. VAE uses a fixed prior with a variable posterior. By contrast, RaDOGAGA uses a parametric prior and posterior with constant variance, which makes the latent space isometric to the input space. This relation is described in Figure 1. So we add the description to clarify the difference at the end of section 2. We hope this description will help the readers to understand RaDOGAGA.
>
> To show our claim more clearly, we clarified our motivation and claims in the introduction such that VAE can be regarded as a rate-distortion optimized transform coding and can be thoroughly understood the quantitative behavior.
>
> Furthermore, we also move the important discussion of the relation with prior works from appendix A to the newly added section 3.4 in the main text.  In the section, we show that VAE objective becomes a log-likelihood of the input. That is, VAE has already achieved the information bottleneck (Tishby et al. 1999) as it is. As a result, the theoretical analysis "ELBO is broken" in Alemi et al. (2018) is incorrect (please see below). We also show what beta-VAE does is only to scale the metrics by a factor of beta from the original VAE.
>
> Regarding the model capacity, we also discussed the detail in the original Appendix A.4.  Our paper shows that beta-VAE can be regarded as a rate-distortion optimized orthonormal transform coding with constant noise beta/2 in each channel.  In this case, the entropy of noise is H(D)=(n/2) \log(beta pi e).  Because the entropy of the input data is H(X), the model capacity, i.e., Rate, is H(X)- H(D) = H(X) - (n/2) \log(beta pi e). This is, VAE latent representation (as it is) achieves the information bottleneck (Tishby et al. 1999) in the rate-distortion sense.   In this sense, the claim "ELBO is broken"   (Alemi et al. 2018) is clearly incorrect.
>
> Regarding the approximations of D(x,\hat x), we add the description that "We show the exhaustive and quantitative evaluation of coding loss and transform loss in the toy dataset in Appendix E.2 to validate this approximation".  No new data is added, but the clarification only.
> Regarding the approximations of KL-divergence, we clarify the condition that our approximation is reasonable when sigma^2<0.1.  If a dimensional component captures meaningful information, a rate -\log(sigma^2) should have reasonable information, otherwise, such VAE does not capture the representation. So sigma^2<0.1 is a condition for a reasonable representation to capture in VAE.  Currently, this explanation is described in the main text as "Thus, in the dimension with meaningful information,σj(x)^2 is much smaller than the prior variance 1".
>
> Please find that we add some explanations to help to understand RaDOGAGA and some expansions in the revision. Although we referred to the rate-distortion theory, Wiener filter, RaDOGAGA, and isometric embedding, we have carefully described all of the expansions and proofs such that it is possible to follow an equation-by-equation basis without the knowledge of related works. We believe our proofs, some details of which are also in the appendix, are self-contained.
>
> As you point out, Eq.8 is incorrect. We revise it to the absolute value of the Jacobian determinant. Because the derivative of \log |x| is 1/x, the expansions after Eq.9 do not change.
>
> We believe that our paper is the culmination of the VAE related theoretic studies.  That is, prior theoretic VAE works discussed in Appendix A can be clearly understood based on our analysis.
>
> Furthermore, our analysis will spur the information-theoretic / rate-distortion based generative model.
> For example, there is a problem that VAE can not capture the features well.  Based on our work, VAE can be understood as a PCA/transform coding like lossy compression, where the representation in PCA/transform coding is sensitive to the input data's positional alignment.  If bad positional alignment, entropy as well as representation becomes worse. Once our theory is understood, we will easily find a way to overcome such problems.
>
> In addition, our analysis also helps to choose beta for a data set. Let S be a variance of the input signal in each dimension. In our experience, S/(beta/2) \simeq 1000 is a sweet spot in most cases (This is not described in our paper). If you have a chance, please choose beta for several data based on our analysis.
>
> We hope our explanations and revision will help you to understand our paper clearer and deeper.

---

### Official Review · AnonReviewer2 · 2020-10-28
**Review of paper 1118**

**Rating:** 4
**Confidence:** 4

**Review:**

The paper builds on a branch of recent works that consider and analyse Variational autoencoders (VAE) from the view point of data compression. This started from Alemi et al. 2018, where the authors consider and analyse the mutual information between data and latent codes and culminates in Kato et al. 2020, where the authors consider models in which both the encoder and the decoder are assumed as deterministic (isometric) mappings. The submitted paper aims at reconciling this type of models with standard VAEs by claiming to prove that VAEs can be obtained from the former by an non-linear component-wise scaling of the latent space. The authors claim among others that this approach allows to estimate the data probability density from the learned VAE model.

Unfortunately, the paper is not well written. This begins with sloppy wording and imprecise notations. Moreover the authors do not explain precisely enough the concepts used in the main building blocks of their proof and their final composition. Altogether, this makes it impossible (at least for me) to follow the steps of the construction and the related proof. In my opinion, the paper in its present state is not publishable at ICLR because it requires a major revision.

I have tried to read the revised version submitted by the authors. Unfortunately, it is still very hard for me to fully grasp the proposed concept and to follow the derivation steps and the proofs. In my opinion, the paper is not publishable in its present state.

---

> ### Author Response · Authors · 2020-11-22
> **Thank you very much for your valuable comments**
>
> Thank you very much for your review.
>
> Because I would like to answer your question, I appreciate it if you point out more clearly.
>
> To clarify our claim more clearly, we clarified our motivation and claims in the introduction such that VAE can be regarded as a rate-distortion optimized transform coding and can be thoroughly understood the quantitative behavior. We also move the important discussion of the relation with prior works from appendix A to the newly added section 3.4 in the main text. Especially, we point out that what beta-VAE actually does is only to scale the metric in the original VAE, and also that well-referred prior work is incorrect.
>
> The purpose of our paper is to clarify what kind of latent space is captured by VAE.
> To prove that, the expected behavior of VAE is first described in 3.1, and we give the proof in 3.2 and 3.3. The idea of rate-distortion theory, Wiener filter, and isometric embedding are referred to, however, we believe all of the expansion is self-contained, and it is possible to follow an equation-by-equation basis.  Finally, we validate the theory by experiments quantitatively.
>
> To clarify our claim more clearly, I will move the description of the fact which has not been discussed, or even been wrongly discussed, from the appendix to the newly added section 3.4 in the main text.
>
> We hope our revision will help you to understand our points.
> We also appreciate it if you point out which part is sloppy. We sincerely consider your point and we will improve our paper based on your comment.

---

### Author Response · Authors · 2020-11-23
**To all reviewers: We could improve our paper based on valuable comments.**

We thank all reviewers for their valuable and careful comments, and also apologize for our late response.
We consider all of their comments seriously and revise our paper as below.

* Section 1. "Introduction"
  - Clarify what has not been discussed in prior works.
  - Clarify the motivation why Rate-distortion theory is introduced.
  - Brief explanation of isometric embedding.
  - Revise claim 1) to explain the relationship with the rate-distortion optimal.

* Section 2. "Related works"
  - Add a concise description what has not been discussed in the prior works in 2.1.
  - Add a concise description to explain the relation between VAE and RaDOGAGA(Kato et al. 2020) at the end of 2.2.

* Section 3. "Understanding of VAE as a scaled isometric embedding"
  - We add a new section 3.4. to explain the relation with prior works.  These discussions were mainly described in Appendix A, and we moved to 3.4. Outline of 3.4 is as follows:
		- VAE optimal can be considered as a rate-distortion optimal of transform coding with allowing beta/2 in each dimension.  As a result, VAE latent representation can be understood by well-studied rate-distortion theory.
		- What beta-VAE actually does is only to scale the metric in the original VAE.
		- Correct the information-theoretic analysis "ELBO is broken" in well-referred prior work(Alemi et al. 2018).
  - Correct Eq.7 and Eq.8 as pointed out by R1.
  - Move the proofs of new Eq.15 and Eq.16 from the main text to Appendix A.1 for easy reading.
  - Add the description that the approximation of the metric is quantitatively verified in the toy dataset.
  - Revise the introduction of section 3.

*Overall
  - We revise the description "We will show later" as pointed out by R4.
  - We correct some typos(tailor expansion => Taylor expansion).

Although we agree that our description is "Information-dense"  or "wordy", please understand this is a trade-off with self-contained explanations. We carefully describe our equations and proofs are mathematically self-contained such that it will be able to follow the equations without the knowledge of rate-distortion theory, Wiener filter, RaDOGAGA(Kato et al. 2020), and isometric embedding. We are confident that all of the equations are needed to show the quantitative properties of VAE. We revised some parts for easier understanding. We hope this will help you understand our paper deeper and clearer.

By this revision, we believe that the behavior of VAE optimal is thoroughly clarified and that prior theoretic VAE works discussed in Appendix A can be clearly understood based on our analysis.  Our work will further spur the information-theoretic generative model.

We hope our explanations and revision will help you to understand our claims clearer and deeper.
Although our response is late, we would appreciate it if you consider our clarification and give us further valuable comments.

---

### Decision · Program_Chairs · 2021-01-07
**Final Decision**

**Decision:**

Reject

**Comment:**

This submission analyses the VAE objective from the perspective of non-linearly scaled isometric embeddings, with the aim of improving our information-theoretic understanding of the variational objective.

Reviewers are in consensus that this submission in its current form is very difficult to read, even after revisions by the authors. The metareviewer, who is highly familiar with information-theoretic and even information-geometric interpretations of VAEs, similarly struggled to understand this paper. Many concepts (e.g. KLT transforms) are not introduced in a self-contained manner, nor are related works like RaDOGAGA. Moreover the exposition introduces lots of notation (often somewhat implicitly) and requires more  high-level plain-English statements that signal the structure of the overall narrative to the reader. As a result, it is hard to understand the paper, even at the level of the contributions that are claimed by the authors. It appears that Section 3.4 should be read as culminating in a "correction" of the rate-distortion view of VAEs proposed by Alemi et al. Unfortunately the metareviewer is not able to understand from the writing what fault the authors find with the proposed interpretation, and how their view informs a better perspective.

It is difficult to provide the authors with concrete addressable suggestions at this stage of revision of their manuscript. The metareviewer's advice would be to attempt to focus on defining a narrative structure that clearly explains what insights this perspective of VAEs contributes, what misconception it corrects, and how it corrects it –– and then focus on streamlining notation in a manner that makes it possible to follow along with the exposition more easily.